# MOF-BFN: Metal-Organic Frameworks Structure Prediction via Bayesian Flow Networks

**Rui Jiao**[1,2,†] **Hanlin Wu**[2,†] **Wenbing Huang**[3,4,*] **Yuxuan Song**[2] **Yawen Ouyang**[2]

**Yu Rong**[5] **Tingyang Xu**[5] **Pengju Wang**[5] **Hao Zhou**[2,*]

**Wei-Ying Ma**[2] **Jingjing Liu**[2] **Yang Liu**[1,2,*]

[1]Dept. of Comp. Sci. & Tech., Institute for AI, Tsinghua University
[2]Institute of AI Industry Research (AIR), Tsinghua University
[3]Gaoling School of Artificial Intelligence, Renmin University of China
[4] Beijing Key Laboratory of Big Data Management and Analysis Methods, Beijing, China
[5] Alibaba DAMO Lab

## Abstract

Metal-Organic Frameworks (MOFs) have attracted considerable attention due to their unique properties including high surface area and tunable porosity, and promising applications in catalysis, gas storage, and drug delivery. Structure prediction for MOFs is a challenging task, as these frameworks are intrinsically periodic and hierarchically organized, where the entire structure is assembled from building blocks like metal nodes and organic linkers. To address this, we introduce MOF-BFN, a novel generative model for MOF structure prediction based on Bayesian Flow Networks (BFNs). Given the local geometry of building blocks, MOF-BFN jointly predicts the lattice parameters, as well as the positions and orientations of all building blocks within the unit cell. In particular, the positions are modelled in the fractional coordinate system to naturally incorporate the periodicity. Meanwhile, the orientations are modeled as unit quaternions sampled from learned Bingham distributions via the proposed Bingham BFN, enabling effective orientation generation on the 4D unit hypersphere. Experimental results demonstrate that MOF-BFN achieves state-of-the-art performance across multiple tasks, including structure prediction, geometric property evaluation, and de novo generation, offering a promising tool for designing complex MOF materials.

## 1 Introduction

Metal-Organic Frameworks (MOFs) have attracted significant interest in recent years due to their unique structural properties and wide range of potential applications [1, 12]. These materials are characterized by high surface areas, tunable porosities, and exceptional versatility, which make them ideal candidates for use in catalysis [9], gas storage [16], drug delivery [14], and other fields [18, 3]. Structurally, MOFs are composed of metal ions or clusters coordinated with organic linkers, forming periodic structures that can be tailored for specific functions [21]. This tunability has led to their

---

*Wenbing Huang, Hao Zhou and Yang Liu are corresponding authors. † indicates equal contribution.
This work is done when Rui Jiao works as an intern in Alibaba Group.

exploration in numerous scientific and industrial domains, sparking a need for more efficient and accurate methods for predicting MOF structures.

Predicting the structures of MOFs is a critical problem. Traditional crystal structure prediction methods typically rely on *ab initio* calculations, using optimization algorithms to find local minima on energy landscapes defined by Density Functional Theory (DFT) or machine learning (ML) force fields [30]. However, such approaches are computationally intensive. To overcome this limitation, recent work has explored deep generative models that directly learn the data distribution, thereby bypassing explicit energy optimization. While these models have shown promising results on small-scale inorganic crystals,

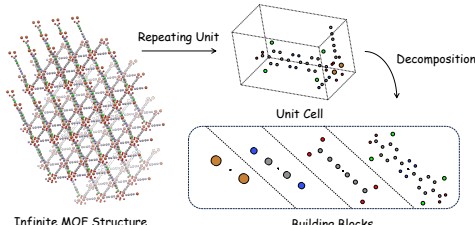

Figure 1: Decomposition of a MOF.

they encounter challenges in addressing MOFs: the unit cells of MOFs often contain hundreds of atoms, making direct atomic-level generation highly complex and computationally expensive. To mitigate this, prior work has adopted a hierarchical modeling paradigm, where MOFs are decomposed into smaller, rigid building blocks [1, 5, 13], as shown in Figure 1. These methods aim to predict the positions and orientations of building blocks to reconstruct the full structure, reducing the complexity of the problem. Particularly, MOFFlow [13] is a representative approach that employs flow-matching techniques to jointly generate both the unit cell parameters and the positions and orientations of each block, offering a promising solution to the structural prediction problem.

Despite its success, MOFFlow represents the unit cell as a finite 3D graph with global features such as lattice parameters, without explicitly modeling the *periodicity* of MOF structures. Periodicity is a key characteristic of crystalline materials, and its proper modeling is essential capturing multiple properties. To this end, recent works have adopted *fractional coordinate systems*, which models atomic positions on a 3D torus normalized by the unit cell, naturally incorporating periodic boundary conditions. Several generative methods based on diffusion [10, 17] or flow matching [20] have been developed to model these fractional coordinates, showcasing promising results in various crystal generation tasks.

However, diffusion- and flow-based models typically rely on continuous-time stochastic differential equations (SDEs) or ordinary differential equations (ODEs), which would inevitably be discretized during sampling. This discretization introduces truncation errors, which can accumulate and degrade model performance [19, 31]. In contrast, Bayesian Flow Networks (BFNs) [6], a recent advancement in generative modeling, reformulate the generation process via iterative Bayesian updates. This paradigm bypasses the need for solving SDEs or ODEs, thus eliminating the discretization errors. Notably, CrysBFN [28], a recent work based on BFNs, has demonstrated strong performance in modeling fractional coordinates for inorganic crystal generation, highlighting the potential of BFN-based frameworks.

Nevertheless, extending BFNs to MOF structure prediction faces additional challenges. In hierarchical MOF representations, building blocks are not simply treated as point particles but as rigid bodies with local geometries [13]. Therefore, in addition to predicting the fractional coordinates of each block, it is necessary to model its orientation in three-dimensional space. Specifically, this requires learning distributions over rotation matrices, or equivalently, over the special orthogonal group $SO(3)$, which remains unsolved in prior BFN-based methods.

To address this gap, we propose MOF-BFN, a novel generative framework for hierarchical MOF structure prediction based on Bayesian Flow Networks. MOF-BFN operates in the fractional coordinate system to preserve periodicity and models block orientations using unit quaternions sampled from Bingham BFNs—a new generative module that extends BFNs to the hypersphere $S^3$ for rotation modeling. Given the geometry of each building block, MOF-BFN jointly predicts the lattice parameters, fractional coordinates, and orientations of all building blocks within a unit cell.

In summary, our contributions are as follows:

- We introduce MOF-BFN, the first hierarchical structure prediction framework that jointly models periodicity, position, and orientation using Bayesian Flow Networks.

- We incorporate fractional coordinates to capture periodicity and employ Bingham distributions to generate orientations in the unit quaternion space.

- We demonstrate the superior performance of MOF-BFN over existing methods in multiple tasks including structure prediction, geometric property evaluation and de novo generation.

## 2 Related Works

**Generative Models for Crystalline Materials.** Generative models for crystalline materials, including both inorganic crystals and metal-organic frameworks (MOFs), have made notable advancements. CDVAE [29] integrates a diffusion decoder into a VAE to generate structures from predicted lattices. DiffCSP [10] improves this by jointly diffusing lattice matrices and fractional coordinates. FlowMM [20] and CrysBFN [28] further improve the generation performance via more advanced generative models for the torus space. FlowLLM [26] initializes flow matching process with a LLM-based prior [7]. TGDMat [4] innovatively introduces text conditions to the generative model. DiffCSP++ [11] considers space group-based generation given specific Wyckoff position (WP) assignments, and SymmCD [15] extends this by further enabling the generation of WPs. Specific for MOFs, MOFDiff [5] extends CDVAE into a hierarchical diffusion model using coarse-grained blocks. While MOFDiff requires post-processing to fully optimize block orientations, MOFFlow [13] directly models the orientation of each building block in the flow matching framework, leading to an end-to-end paradigm. However, MOFFlow does not explicitly model the periodicity of MOFs, which we address by incorporating the fractional coordinate system in this work.

**Bayesian Flow Networks.** Bayesian Flow Networks (BFNs, Graves et al. [6]) define a generative process that transitions from a prior to a posterior through Bayesian updates. Their smooth generation has been effective in 3D tasks like molecule generation [25] and structure-based drug design [23]. CrysBFN [28] extends BFN to torus space via the von Mises distribution. In this work, we further enhance BFN with a hierarchical MOF-specific representation, enabling accurate modeling of both block positions and orientations.

## 3 Preliminaries

### 3.1 Representation of MOFs

**General Representations of MOFs.** An MOF structure can be described as an infinite periodic arrangement of atoms in 3D space, where the smallest repeating unit is called the unit cell. A unit cell containing $N$ atoms can be formed by a triplet $\mathcal{M} = (\boldsymbol{L}, \boldsymbol{X}, \boldsymbol{A})$, where $\boldsymbol{L} = [\boldsymbol{l}_1, \boldsymbol{l}_2, \boldsymbol{l}_3] \in \mathbb{R}^{3\times3}$ is the lattice matrix determining the periodicity of the unit cell, $\boldsymbol{X} = [\boldsymbol{x}_i]_{i=1}^N \in \mathbb{R}^{3\times N}$ is the Cartesian coordinate matrix specifying the positions of the atoms, and $\boldsymbol{A} = [\boldsymbol{a}_i]_{i=1}^N \in \mathbb{R}^{h\times N}$ is the $h$-dimensional one-hot encoding matrix representing the atom types. To describe the shape of the parallelpiped $\boldsymbol{L}$, a commonly-used alternate is the invariant lattice parameter $\boldsymbol{\xi} = (a, b, c, \alpha, \beta, \gamma) \in \mathbb{R}_+^3 \times (0, \pi)^3$ that characterizes the lengths and pairwise angles of the three basis vectors. [2] The entire periodic structure can be formulated by $\{(\boldsymbol{a}_i', \boldsymbol{x}_i') | \boldsymbol{a}_i' = \boldsymbol{a}_i, \boldsymbol{x}_i' = \boldsymbol{x}_i + \boldsymbol{L}\boldsymbol{t}, \forall \boldsymbol{t} \in \mathbb{Z}^{3\times1}\}$, where the integer vector $\boldsymbol{t}$ indicates the arbitrary translation along the basis vectors according to periodicity.

**Block-level Representations of MOFs.** MOFs typically consist of hundreds to thousands of atoms per unit cell, making it challenging to directly design full-atom generative models. To overcome this problem, a hierarchical modeling approach is commonly employed, where atoms are grouped into building blocks such as metal clusters and organic linkers [5, 13]. Formally, a unit cell with $K$ building blocks ($K \ll N$) is then described as $\mathcal{M}^B = (\boldsymbol{L}, \boldsymbol{X}^B, \mathsf{R}^B, \mathcal{B})$, where $\boldsymbol{X}^B \in \mathbb{R}^{3\times K}$ and $\mathsf{R}^B \in \mathrm{SO}(3)^K$ specify the center positions and orientations, $\mathcal{B} = [\mathcal{C}_j]_{j=1}^K$ contains the local structures of blocks. Each block $\mathcal{C}_j = \{(\boldsymbol{a}_r, \boldsymbol{x}_r)\}_{r=1}^{N_j}$ contains $N_j$ atoms, described by atom types $[\boldsymbol{a}_r]_{r=1}^{N_j}$ and local coordinates $[\boldsymbol{x}_r]_{r=1}^{N_j}$. The canonical local coordinates are determined by PCA [13], which is detailed in Appendix C.1. To intrinsically model the periodicity, we further apply the fractional coordinate system, which takes the lattice matrix $\boldsymbol{L}$ as the coordinate basis and describes the relative position of each atom within the unit cell. Specifically, the coordinates are transformed

---

[2]In practice, following Lin et al. [17], we use $\left(\log(a), \log(b), \log(c), \tan(\alpha - \frac{\pi}{2}), \tan(\beta - \frac{\pi}{2}), \tan(\gamma - \frac{\pi}{2})\right)$ to project the parameters in continuous space $\mathbb{R}^6$ to simplify modelling. Hereinafter, the notation $\boldsymbol{\xi}$ denotes this projected representation.

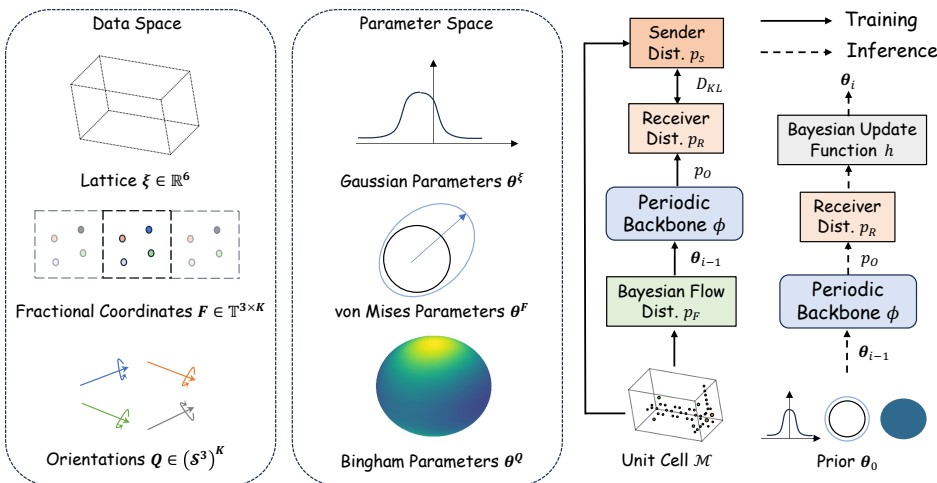

Figure 2: Overview of MOF-BFN. To model the joint distribution over different manifolds in the data space, MOF-BFN defines a generative process in the corresponding parameter space. During training, the model takes $\boldsymbol{\theta}_{i-1}$ from Bayesian flow distribution on the unit cell $\mathcal{M}$ as the input, and outputs an approximation of $\mathcal{M}$. The training objective is to minimize the discrepancy between the estimation and the ground truth. During inference, the process begins from uninformative prior $\boldsymbol{\theta}_0$, and progressively updates the parameters via the trained model.

into $\boldsymbol{F}^B = [\boldsymbol{f}_j^B]_{j=1}^K = [\boldsymbol{L}^{-1}\boldsymbol{x}_j^B]_{j=1}^K \in [0,1)^{3 \times K}$. Hereinafter, we simplify $\boldsymbol{F}^B$ as $\boldsymbol{F}$ to represent the block-level fractional coordinates.

**Represent Rotations as Unit Quaternions.** Apart from the rotation matrices, unit quaternions offer a compact and efficient way to represent 3D rotations. For $\boldsymbol{q} = [w, x, y, z]$ and $\|\boldsymbol{q}\| = 1$, the corresponding rotation matrix is given by

$$\boldsymbol{R}(\boldsymbol{q}) = \begin{pmatrix} 1 - 2(y^2 + z^2) & 2(xy - wz) & 2(xz + wy) \\ 2(xy + wz) & 1 - 2(x^2 + z^2) & 2(yz - wx) \\ 2(xz - wy) & 2(yz + wx) & 1 - 2(x^2 + y^2) \end{pmatrix}. \tag{1}$$

**Task Definition.** The task of MOF structure prediction is to predict the lattice matrix and detailed block arrangement given the block types. By representing lattices as the invariant parameters $\boldsymbol{\xi}$ and encoding orientations as quaternions $\boldsymbol{Q} = [\boldsymbol{q}_i]_{i=1}^K$, the structure prediction task is formulated as capturing the joint distribution $p(\boldsymbol{\xi}, \boldsymbol{F}, \boldsymbol{Q}|\mathcal{B})$. Furthermore, to enable de novo generation, we extend the formulation to model $p(\boldsymbol{\xi}, \boldsymbol{F}, \boldsymbol{Q}, \mathcal{B})$, thereby incorporating the block type generation into the structure prediction framework.

### 3.2 Bayesian Flow Networks

Bayesian Flow Networks (BFNs) [6] are a class of generative models that progressively refine the parameters of a distribution set via Bayesian updates. Specifically, a series of *sender distributions* is constructed by perturbing data $\boldsymbol{x}$ with noise levels corresponding to a predefined accuracy schedule $[\alpha_i]_{i=1}^T$, resulting in $p_S(\boldsymbol{y}_i|\boldsymbol{x}; \alpha_i)$. In parallel, the receiver begins with an uninformative prior distribution parameterized by $\boldsymbol{\theta}_0$, and progressively updates its belief based on the observed noisy samples $[\boldsymbol{y}_i]_{i=1}^T$. At each step $i$, the receiver treats its distribution from the previous step as the *input distribution* $p_I(\boldsymbol{x}|\boldsymbol{\theta}_{i-1})$, and updates it via Bayes' rule as

$$p_I(\boldsymbol{x}|\boldsymbol{\theta}_i) \propto p_I(\boldsymbol{x}|\boldsymbol{\theta}_{i-1})p_S(\boldsymbol{y}_i|\boldsymbol{x}; \alpha_i), \tag{2}$$

which corresponds to the following update rule in the parameter space:

$$\boldsymbol{\theta}_i = h(\boldsymbol{\theta}_{i-1}, \boldsymbol{y}_i, \alpha_i), \tag{3}$$

where $h$ is called the *Bayesian update function*. The corresponding *Bayesian update distribution* is yielded by marginalizing $\boldsymbol{y}_i$

$$p_U(\boldsymbol{\theta}_i|\boldsymbol{\theta}_{i-1}, \boldsymbol{x}; \alpha_i) = \mathbb{E}_{\boldsymbol{y}_i \sim p_S(\boldsymbol{y}_i|\boldsymbol{x}; \alpha_i)}\delta(\boldsymbol{\theta}_i - h(\boldsymbol{\theta}_{i-1}, \boldsymbol{y}_i, \alpha_i)). \tag{4}$$

At step $i$, the marginalized *Bayesian flow distribution* is accumulated from the prior $\boldsymbol{\theta}_0$ as

$$p_F(\boldsymbol{\theta}_i|\boldsymbol{x}, [\alpha_j]_{j=1}^i) = \mathbb{E}_{p_U(\boldsymbol{\theta}_1|\boldsymbol{\theta}_0, \boldsymbol{x}; \alpha_1)} \cdots E_{p_U(\boldsymbol{\theta}_{i-1}|\boldsymbol{\theta}_{i-2}, \boldsymbol{x}; \alpha_{i-1})} p_U(\boldsymbol{\theta}_i|\boldsymbol{\theta}_{i-1}, \boldsymbol{x}; \alpha_i). \tag{5}$$

However, during generation, the sender distributions are actually unavailable as they requires ground-truth data. To address this, a neural network $\phi$ is introduced to approximate the sender distributions by producing *output distributions* that aim to recover the clean data. At each step $i$, the model takes the current parameter $\boldsymbol{\theta}_{i-1}$ as the input and outputs an estimated distribution $p_O(\hat{\boldsymbol{x}}|\phi(\boldsymbol{\theta}_{i-1}, t_i))$, usually selected as the Dirac distribution on the model prediction that

$$p_O(\hat{\boldsymbol{x}}|\phi(\boldsymbol{\theta}_{i-1}, t_i)) = \delta(\hat{\boldsymbol{x}} - \phi(\boldsymbol{\theta}_{i-1}, t_i)). \tag{6}$$

The approximated sender distribution, or so-called *receiver distribution*, is further given by

$$p_R(\boldsymbol{y}_i|\boldsymbol{\theta}_{i-1}; t_i, \alpha_i) = \mathbb{E}_{p_O(\hat{\boldsymbol{x}}|\phi(\boldsymbol{\theta}_{i-1}, t_i))} p_S(\boldsymbol{y}_i|\hat{\boldsymbol{x}}; \alpha_i). \tag{7}$$

The training objective is to minimize the transmission error, quantified as the KL-divergence between the sender and receiver distribution as

$$\mathcal{L}(\phi) = \mathbb{E}_{\boldsymbol{x} \sim p_{\text{data}}, i \sim \mathcal{U}(1,T)} D_{KL}(p_S(\cdot|\boldsymbol{x}; \alpha_i) \| p_R(\cdot|\boldsymbol{\theta}_{i-1}; t_i, \alpha_i)), \tag{8}$$

which is generally measured by the distance or similarity between the predicted and ground truth data. Overall, the construction of BFN requires the following key components: (1) a prior parameter $\boldsymbol{\theta}_0$ for initialization; (2) a Bayesian update function $h$ for progressively refining the parameter; (3) a Bayesian flow distribution $p_F$ accumulated via $h$ for training; and (4) a closed-form KL divergence as the training objective, which will be detailed in the following section.

## 4 MOF-BFN

As illustrated in Fig. 2, to model the joint distribution $p(\boldsymbol{\xi}, \boldsymbol{F}, \boldsymbol{Q}|\mathcal{B})$, we design MOF-BFN over the parameter space of $\boldsymbol{\theta}^{\mathcal{M}} = \{\boldsymbol{\theta}^{\boldsymbol{\xi}}, \boldsymbol{\theta}^{\boldsymbol{F}}, \boldsymbol{\theta}^{\boldsymbol{Q}}\}$. During training, the model samples discrete timestep $i \sim U(1, T)$ and takes $\boldsymbol{\theta}_{i-1}^{\mathcal{M}}$ as the input parameter, and minimize the training objective abstracted as Eq. (8) on each manifold. Initialized from the prior $\boldsymbol{\theta}_0^{\mathcal{M}}$, each inference step first yields the output distribution from the trained model to approximate the receiver distribution, and then refines the input parameters via Bayesian update. The final prediction is drawn from the output distribution at the last step $T$. In this section, we first briefly introduce the design of BFNs on continuous and torus space in § 4.1, and detail the proposed Bingham BFN in § 4.2. Finally, we provide the training schemes of MOF-BFN on the structure prediction task and the extended de novo generation task in § 4.3.

### 4.1 Bayesian Flow Networks on Common Manifolds

**Lattice Parameters on Continuous Space $\mathbb{R}^6$** The input distribution of $\boldsymbol{\xi}$ follows the Gaussian form $\mathcal{N}(\boldsymbol{\xi}; \boldsymbol{\mu}_i^{\boldsymbol{\xi}}, (\rho_i^{\boldsymbol{\xi}})^{-1}\boldsymbol{I})$ parameterized by $\boldsymbol{\theta}_i^{\boldsymbol{\xi}} = \{\boldsymbol{\mu}_i^{\boldsymbol{\xi}}, \rho_i^{\boldsymbol{\xi}}\}$. Following the standard BFN on continuous space [6], the prior distribution is selected as the standard Gaussian with $\boldsymbol{\theta}_0^{\boldsymbol{\xi}} = \{\boldsymbol{0}, 1\}$. After observing a sample from the Gaussian form sender distribution $\boldsymbol{y}_i^{\boldsymbol{\xi}} \sim \mathcal{N}(\boldsymbol{\xi}, (\alpha_i^{\boldsymbol{\xi}})^{-1}\boldsymbol{I})$ at step $i$, the Bayesian update function is given by

$$\{\boldsymbol{\mu}_i^{\boldsymbol{\xi}}, \rho_i^{\boldsymbol{\xi}}\} = \Big\{ \frac{\rho_{i-1}^{\boldsymbol{\xi}}\boldsymbol{\mu}_{i-1}^{\boldsymbol{\xi}} + \alpha_i^{\boldsymbol{\xi}}\boldsymbol{y}_i^{\boldsymbol{\xi}}}{\rho_{i-1}^{\boldsymbol{\xi}} + \alpha_i^{\boldsymbol{\xi}}}, \rho_{i-1}^{\boldsymbol{\xi}} + \alpha_i^{\boldsymbol{\xi}} \Big\}. \tag{9}$$

Leveraging the additive property of Gaussian distributions, given $\alpha_i^{\boldsymbol{\xi}} = \sigma_T^{-2i/T}(1 - \sigma_T^{2/T})$, and $\sigma_T$ is a predefined small variance, the Bayesian flow distribution can be analytically obtained as

$$p_F^{\boldsymbol{\xi}}(\boldsymbol{\mu}_i^{\boldsymbol{\xi}}|\boldsymbol{\xi}, i) = \mathcal{N}\big((1 - \sigma_T^{2i/T})\boldsymbol{\xi}, \sigma_T^{2i/T}(1 - \sigma_T^{2i/T})\boldsymbol{I}\big), \tag{10}$$

with $\rho_i^{\boldsymbol{\xi}} = \sigma_T^{-2i/T}$. Since both the sender and receiver distributions are Gaussian with identical variance, their KL divergence simplifies to a mean-squared error form as

$$\mathcal{L}_{\boldsymbol{\xi}} = \mathbb{E}_{i \sim U(1,T), \boldsymbol{\mu}_{i-1}^{\boldsymbol{\xi}} \sim p_F^{\boldsymbol{\xi}}(\boldsymbol{\mu}_{i-1}^{\boldsymbol{\xi}}|\boldsymbol{\xi}, i-1)} \Big[ \frac{\alpha_i^{\boldsymbol{\xi}}T}{2} \|\boldsymbol{\xi} - \phi_{\boldsymbol{\xi}}(\boldsymbol{\theta}_{i-1}^{\mathcal{M}}, i)\|_2^2 \Big]. \tag{11}$$

**Fractional Coordinates on Torus** $\mathbb{T}^{3 \times K}$  As established by Wu et al. [28], circular distributions are appropriate for modeling the periodicity of fractional coordinates, and one suitable choice is the von Mises distribution defined as $p_v(x; m, \kappa) \propto \exp(\kappa \cos(2\pi(x - m)))$ with $x, m \in [0, 1)$ denote the random variable and mean direction, and $\kappa \geq 0$ controls the concentration. Parameterized as $\boldsymbol{\theta}_i^{\boldsymbol{F}} = \{\boldsymbol{m}_i^{\boldsymbol{F}}, \boldsymbol{\kappa}_i^{\boldsymbol{F}}\}$, the input distribution is initialized by $\boldsymbol{m}_0^{\boldsymbol{F}} \sim U(0, 1), \boldsymbol{\kappa}_0^{\boldsymbol{F}} = \boldsymbol{0}$. We can further describe the parameter in the complex form with $\boldsymbol{c}_i^{\boldsymbol{F}} = \boldsymbol{\kappa}_i^{\boldsymbol{F}} e^{2\pi \boldsymbol{m}_0^{\boldsymbol{F}} \boldsymbol{i}}$ where $\boldsymbol{i}$ is the imaginary unit, implying that $\boldsymbol{\kappa}_i^{\boldsymbol{F}}, \boldsymbol{m}_i^{\boldsymbol{F}}$ depict the modulus and argument of $\boldsymbol{c}_i^{\boldsymbol{F}}$. Acquiring the sample $\boldsymbol{y}_i^{\boldsymbol{F}} \sim p_v(p_v(\boldsymbol{F}, \alpha_i^{\boldsymbol{F}}))$ with pre-defined $\alpha_i^{\boldsymbol{F}}$, the Bayesian update function at step $i$ is defined as

$$\boldsymbol{c}_i^{\boldsymbol{F}} = \boldsymbol{c}_{i-1}^{\boldsymbol{F}} + \alpha_i^{\boldsymbol{F}} e^{2\pi \boldsymbol{y}_i^{\boldsymbol{F}} \boldsymbol{i}}, \tag{12}$$

resulting in the accumulated Bayesian flow distribution as

$$p_F^{\boldsymbol{F}}(\boldsymbol{c}_i^{\boldsymbol{F}} | \boldsymbol{F}, [\alpha_j^{\boldsymbol{F}}]_{j=1}^i) = \mathbb{E}_{p_v(\boldsymbol{y}_1^{\boldsymbol{F}} | \boldsymbol{F}, \alpha_1^{\boldsymbol{F}}) \cdots p_v(\boldsymbol{y}_i^{\boldsymbol{F}} | \boldsymbol{F}, \alpha_i^{\boldsymbol{F}})} \delta\Big(\boldsymbol{c}_i^{\boldsymbol{F}} - \sum_{j=1}^i \alpha_j^{\boldsymbol{F}} e^{2\pi \boldsymbol{y}_j^{\boldsymbol{F}} \boldsymbol{i}}\Big). \tag{13}$$

The training objective is the KL-divergence between von Mises distributions as

$$\mathcal{L}_{\boldsymbol{F}} = \mathbb{E}_{i \sim U(1,T), \boldsymbol{c}_{i-1}^{\boldsymbol{F}} \sim p_F^{\boldsymbol{F}}(\boldsymbol{c}_i^{\boldsymbol{F}} | \boldsymbol{F}, [\alpha_j^{\boldsymbol{F}}]_{j=1}^{i-1})} \Big[ \alpha_i^{\boldsymbol{F}} T \frac{I_0(\alpha_i^{\boldsymbol{F}})}{I_1(\alpha_i^{\boldsymbol{F}})} \Big( 1 - \cos\big(2\pi(\boldsymbol{F} - \phi_{\boldsymbol{F}}(\boldsymbol{\theta}_{i-1}^{\mathcal{M}}, i))\big)\Big)\Big], \tag{14}$$

where $I_0(\cdot), I_1(\cdot)$ are the modified Bessel functions.

### 4.2  Bingham BFN for Quaternion Generation

In this section, we detail the design of key components for implementing a BFN in the quaternion parameter space $\boldsymbol{\theta}^Q$. For simplicity, we omit the superscript $Q$ and focus on a single quaternion $\boldsymbol{q} \in \mathcal{S}$; the full design for $\boldsymbol{Q} = [\boldsymbol{q}_i]_{i=1}^K$ follows naturally by extension.

**Bingham Distribution.** To model variation in 3D orientations, one often requires a probability distribution defined over the space of rotations. Since each 3D rotation can be represented by a unit quaternion on $\mathcal{S}^3$, it is natural to consider distributions supported on this manifold. However, one key point according to Eq. (1) is that the mapping from unit quaternions to rotation matrices is two-to-one, *i.e.* $R(\boldsymbol{q}) = R(-\boldsymbol{q})$. Hence, any valid distribution must respect this antipodal symmetry. Fortunately, a directional distribution on the hypersphere called *Bingham distribution* serves precisely this role. Mathematically, the Probability Density Function (PDF) of the Bingham distribution takes the form

$$p_B(\boldsymbol{q}; \boldsymbol{M}, \boldsymbol{\Lambda}) = \frac{1}{Z(\boldsymbol{\Lambda})} \exp\big(\boldsymbol{q}^\top \boldsymbol{M}^\top \boldsymbol{\Lambda} \boldsymbol{M} \boldsymbol{q}\big), \tag{15}$$

where $Z(\cdot)$ is the normalization term, and $\boldsymbol{M}^\top \boldsymbol{\Lambda} \boldsymbol{M}$ is the eigendecomposition of a symmetric matrix with orthogonal $\boldsymbol{M} \in \mathbb{R}^{4 \times 4}$ as the eigenvectors, and diagonal $\boldsymbol{\Lambda}$ as the eigenvalues. Based on Eq. (15), one can directly derive the following properties of the Bingham distribution:

**Proposition 1.** *The PDF of the Bingham distribution maintains the antipodal symmetry,* i.e., $p_B(\boldsymbol{q}; \boldsymbol{M}, \boldsymbol{\Lambda}) = p_B(-\boldsymbol{q}; \boldsymbol{M}, \boldsymbol{\Lambda})$.

**Proposition 2.** *When $\boldsymbol{\Lambda} = \boldsymbol{0}$, the PDF is reduced to a uniform distribution,* i.e., $p_B(\boldsymbol{q}; \boldsymbol{M}, \boldsymbol{0}) \equiv \frac{1}{2\pi^2}$.

**Proposition 3.** *Due to the normalization constraint on the hypersphere, any bias applied on the eigenvalues would not affect the distribution,* i.e., $p_B(\boldsymbol{q}; \boldsymbol{M}, \boldsymbol{\Lambda} + k\boldsymbol{I}) = p_B(\boldsymbol{q}; \boldsymbol{M}, \boldsymbol{\Lambda}), \forall k \in \mathbb{R}$.

According to proposition 3, $\boldsymbol{\Lambda}$ is typically reduced to $\boldsymbol{\Lambda} = \text{diag}(\boldsymbol{\lambda}) = \text{diag}(0, \lambda_1, \lambda_2, \lambda_3)$ with $\lambda_3 \leq \lambda_2 \leq \lambda_1 \leq 0$. Hence, a Bingham distribution can also be parameterized by $\boldsymbol{\theta} = \{\boldsymbol{M}, \boldsymbol{\lambda}\}$. By proposition 2, we initialize the parameters using the uniform prior $\boldsymbol{\theta}_0 = \{\boldsymbol{M}_0, \boldsymbol{0}\}$, where $\boldsymbol{M}_0$ is an arbitrary orthogonal matrix[3].

**Bayesian Update of Bingham Distribution.** In Bayesian statistics, a *conjugate prior* refers to a prior distribution that, when combined with a specific form of *likelihood function*, yields a posterior

---

[3]Theoretically, when the Bingham concentration matrix is initialized as $\boldsymbol{\Lambda} = \boldsymbol{0}$, the matrix $\boldsymbol{A}_0 = \boldsymbol{M}_0^\top \boldsymbol{\Lambda} \boldsymbol{M}$ becomes a zero matrix regardless of the choice of orthogonal matrix $\boldsymbol{M}$, resulting in a uniform distribution over the unit sphere. In practice, we initialize $\boldsymbol{M}$ by performing `torch.linalg.eigh` on the zero matrix, which yields the identity matrix $\boldsymbol{M}_0 = \boldsymbol{I}$.

distribution of the same functional form as the prior. This property is greatly essential for the Bayesian update process defined by Eq. (2-3), where the input distribution and sender distribution act as the conjugate prior and likelihood, respectively. Notably, the likelihood need not share the same form as the prior. For the Bingham distribution, it admits a particularly convenient conjugate relationship when the likelihood is modeled using the *Watson distribution*, which yields the PDF as

$$p_W(\boldsymbol{q}; \boldsymbol{\mu}, \kappa) = C_d(\kappa) \exp\left(\kappa(\boldsymbol{\mu}^\top \boldsymbol{q})^2\right). \tag{16}$$

Here $C_d(\cdot)$ is the normalization term, $\boldsymbol{\mu} \in \mathcal{S}^3$ is the mean direction and $\kappa \geq 0$ determines the concentration. The squared inner product ensures antipodal symmetry, consistent with the Bingham distribution. Given $\boldsymbol{\theta}_{i-1} = \{\boldsymbol{M}_{i-1}, \boldsymbol{\lambda}_{i-1}\}$ at step $i$ and an observation $\boldsymbol{y}_i$ from the sender distribution following $p_W(\boldsymbol{y}_i; \boldsymbol{q}, \alpha_i)$, the posterior is updated as

$$p(\boldsymbol{q}|\boldsymbol{\theta}_{i-1}, \boldsymbol{y}_i, \alpha_i) \propto p_B(\boldsymbol{q}; \boldsymbol{M}_{i-1}, \text{diag}(\boldsymbol{\lambda}_{i-1})) \cdot p_W(\boldsymbol{y}_i; \boldsymbol{q}, \alpha_i) \tag{17}$$

$$\propto \exp\left(\boldsymbol{q}^\top \boldsymbol{M}_{i-1}^\top \text{diag}(\boldsymbol{\lambda}_{i-1})\boldsymbol{M}_{i-1}\boldsymbol{q} + \alpha_i(\boldsymbol{y}_i^\top \boldsymbol{q})^2\right) \tag{18}$$

$$= \exp\left(\boldsymbol{q}^\top (\boldsymbol{M}_{i-1}^\top \text{diag}(\boldsymbol{\lambda}_{i-1})\boldsymbol{M}_{i-1} + \alpha_i \boldsymbol{y}_i \boldsymbol{y}_i^\top)\boldsymbol{q}\right) \tag{19}$$

As Eq. (19) retains the form of a Bingham distribution, the Bayesian update function $\boldsymbol{\theta}_i = \{\boldsymbol{M}_i, \boldsymbol{\lambda}_i\} = h(\boldsymbol{\theta}_{i-1}, \boldsymbol{y}_i, \alpha_i)$ is thus defined as

$$\boldsymbol{M}_i, \ \text{diag}(\boldsymbol{\lambda}_i') = \text{EigenDecomposition}\left(\boldsymbol{M}_{i-1}^\top \text{diag}(\boldsymbol{\lambda}_{i-1})\boldsymbol{M}_{i-1} + \alpha_i \boldsymbol{y}_i \boldsymbol{y}_i^\top\right), \tag{20}$$

$$\boldsymbol{\lambda}_i = \boldsymbol{\lambda}_i' - \max(\boldsymbol{\lambda}_i'). \tag{21}$$

The distribution of the updated parameters is finally acquired as

$$p_U(\boldsymbol{\theta}_i|\boldsymbol{\theta}_{i-1}, \boldsymbol{q}; \alpha_i) = \mathbb{E}_{\boldsymbol{y}_i \sim p_W(\boldsymbol{y}_i; \boldsymbol{q}, \alpha_i)}\delta(\boldsymbol{\theta}_i - h(\boldsymbol{\theta}_{i-1}, \boldsymbol{y}_i, \alpha_i)). \tag{22}$$

**Efficient Sampling for Bayesian Flow Distribution.** For BFNs applied to continuous spaces such as Gaussian distributions over $\mathbb{R}^n$, a key additive property holds as

$$p_U(\boldsymbol{\theta}''|\boldsymbol{\theta}, \boldsymbol{q}; \alpha_a + \alpha_b) = \mathbb{E}_{p_U(\boldsymbol{\theta}'|\boldsymbol{\theta}, \boldsymbol{q}; \alpha_a)}p_U(\boldsymbol{\theta}''|\boldsymbol{\theta}', \boldsymbol{q}; \alpha_b), \tag{23}$$

which allows the cumulative accuracy at step $i$ to be expressed as $\rho_i = \rho_0 + \sum_{j=1}^i \alpha_i$. However, the property does not extend to the Bayesian updates in Eq. (22), as the eigenvalues involved in the Bingham updates are not additive. To address this, we follow the strategy similar to Eq. (13) on the von Mises distribution, which also suffers from non-additive property. We simulate the posterior $\boldsymbol{\theta}_i$ by accumulating the outer products in the exponential term. Given $\boldsymbol{A}_i = \boldsymbol{M}_i^\top \text{diag}(\boldsymbol{\lambda}_i')\boldsymbol{M}_i$, we have

$$p_F\left(\boldsymbol{A}_i|\boldsymbol{q}; [\alpha_j]_{j=1}^i\right) = \mathbb{E}_{p_W(\boldsymbol{y}_1|\boldsymbol{q}, \alpha_1)\cdots p_W(\boldsymbol{y}_i|\boldsymbol{q}, \alpha_i)}\delta\left(\boldsymbol{A}_i - \sum_{j=1}^i \alpha_j \boldsymbol{y}_j \boldsymbol{y}_j^\top\right). \tag{24}$$

The samples $[\boldsymbol{y}_j]_{j=1}^i$ can be drawn independently and in parallel via rejection sampling from the angular central Gaussian (ACG) distribution, as described in Appendix C.2. Finally, the schedule of accuracy $[\alpha_i]_{i=1}^T$ is computed numerically to approximate a linearly decreasing entropy, with further implementation details provided in Appendix C.3.

**Training Objective.** The training objective is computed between the predicted and target Watson distributions, which is defined as

$$\mathcal{L}_{\boldsymbol{q}} = \mathbb{E}_{i \sim U(1,T), \boldsymbol{A}_i \sim p_F(\boldsymbol{A}_{i-1}|\boldsymbol{q}, [\alpha_j]_{j=1}^{i-1})}\left[A_d(\alpha_i)T\left(1 - (\boldsymbol{q}^\top \phi_{\boldsymbol{q}}(\boldsymbol{\theta}_{i-1}^{\mathcal{M}}, i))^2\right)\right], \tag{25}$$

where $A_d(\cdot)$ is the second moment of the Watson distribution. Note that Eq. (25) measures the square of the cosine similarity between the prediction $\phi_{\boldsymbol{q}}(\boldsymbol{\theta}_{i-1}^{\mathcal{M}}, i)$ and the ground truth $\boldsymbol{q}$, maintaining the antipodal symmetry.

### 4.3 Training Scheme

The entire training objective aggregates the losses defined on each manifold, resulting in the following composite loss function:

$$\mathcal{L}_{\text{SP}} = \gamma_{\boldsymbol{\xi}}\mathcal{L}_{\boldsymbol{\xi}} + \gamma_{\boldsymbol{F}}\mathcal{L}_{\boldsymbol{F}} + \gamma_{\boldsymbol{q}}\mathcal{L}_{\boldsymbol{q}}, \tag{26}$$

Table 1: **Structure prediction accuracy.** Results for MOF structure prediction. - indicates no match. stol represents the site tolerance for matching criteria. Baseline results are from Kim et al. [13].

| | # of samples | stol = 0.5 | | stol = 1.0 | |
|---|---|---|---|---|---|
| | | MR (%) ↑ | RMSE ↓ | MR (%) ↑ | RMSE ↓ |
| RS [30] | 20 | 0.00 | - | 0.00 | - |
| EA [30] | 20 | 0.00 | - | 0.00 | - |
| DiffCSP [10] | 1 | 0.09 | 0.3961 | 23.12 | 0.8294 |
| | 5 | 0.34 | 0.3848 | 38.94 | 0.7937 |
| MOFFLOW [13] | 1 | 31.69 | 0.2820 | 87.46 | 0.5183 |
| | 5 | 44.75 | 0.2694 | **100.0** | 0.4645 |
| MOF-BFN | 1 | 35.27 | 0.2735 | 92.99 | 0.5000 |
| | 5 | **53.51** | **0.2498** | 98.37 | **0.4117** |

where each $\lambda$ balances the contribution of the corresponding component.

**Backbone Model for MOF-BFN.** The structure prediction backbone contains two main components. The first is the building block encoder $\mathcal{E}$, which maps the local structures of building blocks into invariant latent vectors. In our implementation, we adopt EGNN [24] as the encoder. The second is the coarse-grained structure predictor adapted from the periodic GNN designed in [10], which takes the timestep $i$, the joint parameter $\boldsymbol{\theta}_i^{\mathcal{M}}$, and the encoded condition $\mathcal{E}(\mathcal{B})$ as the input. Importantly, it also incorporates the accuracy terms $\boldsymbol{\kappa}_i^{\boldsymbol{F}}$ and $\boldsymbol{\lambda}_i$, as they indicates the current entropies of the non-additive distributions [28]. The predictor outputs the estimated lattice parameters, quaternions, and fractional translations, which together define the output distributions at each step.

**Extension to De Novo Generation.** We further extend our method to the de novo generation task, where both the types and arrangements of building blocks are required for generation. To this end, we employ a Gaussian BFN trained on the continuous embedding space learned by the contrastive building block encoder from MOFDiff [5], to determine the identity of each block. The training objective is extended as

$$\mathcal{L}_{\mathrm{DNG}} = \gamma_{\boldsymbol{\xi}}\mathcal{L}_{\boldsymbol{\xi}} + \gamma_{\boldsymbol{F}}\mathcal{L}_{\boldsymbol{F}} + \gamma_{\boldsymbol{q}}\mathcal{L}_{\boldsymbol{q}} + \gamma_{\mathcal{B}}\mathcal{L}_{\mathcal{B}}, \tag{27}$$

where $\mathcal{L}_{\mathcal{B}}$ optimizes the generation process on block embedding space and shares a similar form as Eq. (34). More details are referred to Appendix D.

## 5 Experiments

In this section, we evaluate MOF-BFN across a variety of tasks. In § 5.1, we show that MOF-BFN significantly outperforms existing full-atom and hierarchical approaches in structure prediction accuracy. In § 5.2, we demonstrate that the predicted structures exhibit strong agreement with ground-truth structural properties. In § 5.3, we further extend our method to the de novo generation task, where the identity of each block is also required to be determined. Finally, we provide analyses on the efficiency of the fractional modelling and the BFN-based framework in § 5.4.

### 5.1 Structure Prediction

**Setup** We use the BW-DB dataset of 324,426 MOFs from Boyd et al. [1], and decompose each structure into building blocks using the `metal-oxo` algorithm in `MOFid` [2], following Fu et al. [5]. As suggested by Kim et al. [13], we remove structures with over 200 blocks and split the remaining data into training, validation, and test sets in an 8:1:1 ratio.

**Baselines** We compare against three types of baselines: (1) optimization-based methods including Random Search (RS) and Evolutionary Algorithm (EA) from CrySPY [30]; (2) DiffCSP [10], a full-atom generative model without block-level modeling; and (3) MOFFlow [13], a hierarchical generative model using flow matching on block representations.

**Evaluation Metrics** For each test structure, we generate $k$ candidates and report match rate (MR) and root mean square error (RMSE). MR measures the proportion that at least one of the generated structures matches the ground truth, while RMSE is calculated between the ground truth and the best

matching. Matching is performed using `StructureMatcher` from `pymatgen` [22]. We follow Kim et al. [13] and evaluate under two tolerance settings for site, length and angle tolerance criteria: $(0.5, 0.3, 10.0)$ (strict) and $(1.0, 0.3, 10.0)$ (relaxed).

**Results** Table 1 reports the structure prediction results. As expected, optimization-based methods (RS and EA) completely fail to recover correct structures, achieving 0% match rates, which reflects the inefficiency of atom-level search for complex MOFs. DiffCSP shows slightly better performance but still struggles due to its lack of block-level awareness, confirming the limitations of full-atom generative models for large systems. Compared to these baselines, both MOFFlow and our proposed method, MOF-BFN, achieve significantly higher accuracy. Notably, MOF-BFN consistently outper-

Table 2: **Geometric property evaluation.** RMSE computed between the ground-truth and generated MOFs. Baseline results are from Kim et al. [13].

|  | MOF-BFN | MOFFlow | DiffCSP |
|---|---|---|---|
| VSA ($m^2/cm^3$) | **232.8** | 264.5 | 796.9 |
| GSA ($m^2/g$) | **247.5** | 331.6 | 1561.9 |
| AV ($Å^3$) | **315.4** | 530.5 | 3010.2 |
| UCV ($Å^3$) | **312.0** | 569.5 | 3183.4 |
| VF | **0.0187** | 0.0285 | 0.2167 |
| PLD ($Å$) | **0.9072** | 1.0616 | 4.0581 |
| LCD ($Å$) | **0.9257** | 1.1083 | 4.5180 |
| DST ($g/cm^3$) | **0.0185** | 0.0442 | 0.3711 |

forms MOFFlow under strict and relaxed matching criteria. Under the more realistic threshold of $\mathrm{stol} = 0.5$, MOF-BFN achieves a higher MR (53.51% vs. 44.75%) and lower RMSE (0.2498 vs. 0.2694) when generating 5 samples. Even under the relaxed $\mathrm{stol} = 1.0$ setting, MOF-BFN maintains competitive accuracy with a lower RMSE. These results demonstrate that MOF-BFN improves predictive accuracy.

## 5.2 Geometric Property Evaluation

In this section, we assess whether the predicted MOF structures preserve essential structural properties that are critical for downstream applications such as gas storage and separation. To this end, we evaluate eight commonly used structural descriptors: volumetric surface area (VSA), gravimetric surface area (GSA), accessible volume (AV), unit cell volume (UCV), void fraction (VF), pore limiting diameter (PLD), largest cavity diameter (LCD), and density (DST). For each test case, we generate a single structure and compute its properties using `Zeo++` [27]. The RMSE between the predicted and ground-truth property values is then reported in Table 2.

**Results** Table 2 reveals that MOF-BFN achieves the lowest property prediction error across all evaluated metrics, indicating that the model not only aligns structures accurately but also preserves the spatial characteristics and material properties vital for real-world applications.

## 5.3 De Novo Generation

For de novo generation, we generate 1,000 MOFs from both MOF-BFN and MOFDiff without any relaxation and evaluate them using two sets of metrics in Table 3. First, we assess **connection point matching**, where the number of metal and non-metal connection points, identified at the broken bonds between decomposed building blocks, should be equal. MOF-BFN achieves a substantially higher matching rate (0.923 v.s. 0.723), indicating more accurate topological assembly. Second, we apply the `MOFChecker` tool [8] to the matched structures to evaluate their chemical and structural validity. MOF-BFN consistently outperforms MOFDiff across most criteria and a higher proportion of valid structures passing all checks (0.323 v.s. 0.107). These results demonstrate that MOF-BFN is more effective at generating chemically plausible and topologically consistent MOF structures. More details are provided in Appendix D.

Table 3: **Generation validity.** Validation rates of 1,000 generated samples.

| Validity Criteria | MOF-BFN | MOFDiff |
|---|---|---|
| *Connection Point Matching* | | |
| matched ↑ | **0.923** | 0.723 |
| *MOFChecker* | | |
| has_carbon ↑ | **1.000** | **1.000** |
| has_hydrogen ↑ | 0.975 | **0.989** |
| has_atomic_overlaps ↓ | **0.115** | 0.259 |
| has_overcoordinated_c ↓ | **0.193** | 0.365 |
| has_overcoordinated_n ↓ | 0.049 | **0.047** |
| has_overcoordinated_h ↓ | **0.180** | 0.342 |
| has_undercoordinated_c ↓ | **0.182** | 0.248 |
| has_undercoordinated_n ↓ | 0.154 | **0.126** |
| has_undercoordinated_rare_earth ↓ | **0.000** | **0.000** |
| has_metal ↑ | **1.000** | **1.000** |
| has_lone_molecule ↓ | **0.186** | 0.437 |
| has_high_charges ↓ | **0.069** | 0.144 |
| has_suspicious_terminal_oxo ↓ | **0.000** | 0.001 |
| has_undercoordinated_alkali_alkaline ↓ | 0.027 | **0.001** |
| has_geometrically_exposed_metal ↓ | **0.304** | 0.350 |
| all_checks ↑ | **0.350** | 0.148 |
| *Total* | | |
| total_valid ↑ | **0.323** | 0.107 |

### 5.4 Analyses

Our method improves upon the strong baseline MOFFlow in two key aspects: (1) we replace Cartesian coordinates with fractional coordinates to better capture the inherent periodicity of MOFs; and (2) we redesign the generative modeling paradigm using BFNs in place of the original flow matching approach. To evaluate the individual contribution of each modification, we construct an intermediate model named MOF-FracFlow. This variant also applies the flow matching framework but operates in the fractional coordinate space, using the same periodic backbone architecture as MOF-BFN. A comparison of these models is presented in Figure 3, highlighting two key observations. First, MOF-FracFlow outperforms the original MOFFlow, particularly in terms of the match rate with 5 samples, underscoring the importance of incorporating periodicity. Second, MOF-BFN consistently surpasses MOF-FracFlow, demonstrating that after introducing an additional generative target on the $SO(3)$ manifold, BFNs continue to outperform flow matching counterparts, consistent with prior findings [25, 28]. Additional analyses about orientation representations and hyperparameters are provided in Appendix F.

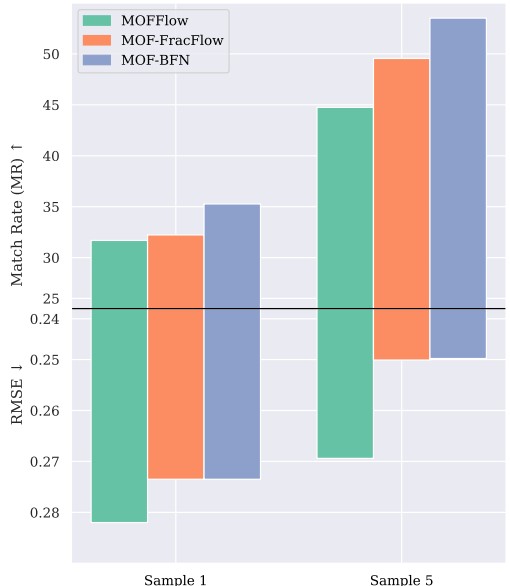

Figure 3: Comparison among different variants. We include MOF-FracFlow as an intermediate model using fractional coordinates system and flow matching for generation.

## 6 Conclusion

In this work, we present MOF-BFN, a hierarchical generative framework for Metal-Organic Frameworks that integrates Bayesian Flow Networks with fractional coordinates and quaternion-based orientation modeling. By operating in the fractional coordinate space and modeling orientation using Bingham distributions over unit quaternions, MOF-BFN jointly captures the periodicity, positions, and orientations of MOF building blocks in a unified generative process. Extensive experiments demonstrate that MOF-BFN achieves state-of-the-art accuracy in structure prediction, property evaluation, and de novo generation, highlighting the effectiveness of MOF-BFN for MOF design.

## Acknowledgments

This work is jointly supported by the National Science and Technology Major Project (No.2022ZD0117502), the National Key R&D Program of China (No.2022ZD0160502), the National Natural Science Foundation of China (No. 61925601, No. 62376276, No. 62276152), and Beijing Nova Program (20230484278).

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

# A Explanations of Key Notations

Table 4: Key notations used in this paper.

| Notation | Meaning / Description |
|---|---|
| $\theta_0$ | Prior parameter for initialization of Bayesian Flow Network (BFN) |
| $p_U$ | Bayesian update for refining parameters iteratively |
| $p_F$ | Bayesian flow distribution accumulated via updates, used for training |
| $\boldsymbol{L} \in \mathbb{R}^{3 \times 3}$ | Lattice matrix |
| $\boldsymbol{\xi} \in \mathbb{R}^6$ | Unconstrained lattice parameters, modeled by Gaussian distribution |
| $\theta_{\xi,i} = \{\mu_{\xi,i}, \rho_{\xi,i}\}$ | Parameters (mean, precision) of Gaussian distribution for $\boldsymbol{\xi}$ at step $i$ |
| $\boldsymbol{F} \in \mathbb{T}^{3 \times K}$ | Block-level fractional coordinates on torus space, modeled by von Mises distribution |
| $\theta_i^F = \{\boldsymbol{m}_i^F, \boldsymbol{\kappa}_i^F\}$ | Parameters of von Mises distribution for $\boldsymbol{F}$ (mean direction and concentration) at step $i$ |
| $\boldsymbol{c}_i^F$ | Complex form representing fractional coordinate parameters |
| $\boldsymbol{Q} \in (\mathcal{S}^3)^K$ | Quaternion representation for 3D orientations |
| $\boldsymbol{q} \in \mathcal{S}^3$ | One single quaternion, modeled by Bingham distribution |
| $\theta_i^q = \{\boldsymbol{M}_i, \boldsymbol{\lambda}_i\}$ | Bingham distribution parameters for quaternions with orientation matrix $\boldsymbol{M}_i$ and normalized eigenvalues $\boldsymbol{\lambda}_i$ at step $i$ |
| $\mathcal{L}$ | Loss functions |
| $\gamma$ | Loss weights |

# B Proofs of Propositions

In this section, we prove the properties listed in § 4.2.

The PDF of the Bingham distribution is rewritten as

$$p_B(\boldsymbol{q}; \boldsymbol{M}, \boldsymbol{\Lambda}) = \frac{1}{Z(\boldsymbol{\Lambda})} \exp\left(\boldsymbol{q}^\top \boldsymbol{M}^\top \boldsymbol{\Lambda} \boldsymbol{M} \boldsymbol{q}\right).$$

**Proposition 1.** *The PDF of the Bingham distribution maintains the antipodal symmetry,* i.e., $p_B(\boldsymbol{q}; \boldsymbol{M}, \boldsymbol{\Lambda}) = p_B(-\boldsymbol{q}; \boldsymbol{M}, \boldsymbol{\Lambda})$.

*Proof.*

$$\begin{aligned}
p_B(-\boldsymbol{q}; \boldsymbol{M}, \boldsymbol{\Lambda}) &= \frac{1}{Z(\boldsymbol{\Lambda})} \exp\left((-\boldsymbol{q}^\top) \boldsymbol{M}^\top \boldsymbol{\Lambda} \boldsymbol{M} (-\boldsymbol{q})\right) \\
&= \frac{1}{Z(\boldsymbol{\Lambda})} \exp\left(\boldsymbol{q}^\top \boldsymbol{M}^\top \boldsymbol{\Lambda} \boldsymbol{M} \boldsymbol{q}\right) \\
&= p_B(\boldsymbol{q}; \boldsymbol{M}, \boldsymbol{\Lambda})
\end{aligned}$$

$\square$

**Proposition 2.** *When $\boldsymbol{\Lambda} = \boldsymbol{0}$, the PDF is reduced to a uniform distribution,* i.e., $p_B(\boldsymbol{q}; \boldsymbol{M}, \boldsymbol{0}) \equiv \frac{1}{2\pi^2}$.

*Proof.* When $\boldsymbol{\Lambda} = \boldsymbol{0}$, the exponent becomes zero for all $\boldsymbol{q} \in \mathcal{S}^3$:

$$p_B(\boldsymbol{q}; \boldsymbol{M}, \boldsymbol{0}) = \frac{1}{Z(\boldsymbol{0})} \exp\left(\boldsymbol{q}^\top \boldsymbol{M}^\top \boldsymbol{0} \boldsymbol{M} \boldsymbol{q}\right) = \frac{1}{Z(\boldsymbol{0})}.$$

That is, $p_B(\boldsymbol{q}; \boldsymbol{M}, \boldsymbol{0})$ is constant over $\mathcal{S}^3$, indicating it is a uniform distribution. Since the surface area of $\mathcal{S}^3$ is $2\pi^2$, the normalized uniform density is $p_B(\boldsymbol{q}; \boldsymbol{M}, \boldsymbol{0}) \equiv \frac{1}{2\pi^2}$. $\square$

**Proposition 3.** *Due to the normalization constraint on the hypersphere, any bias applied on the eigenvalues would not affect the distribution,* i.e., $p_B(\boldsymbol{q}; \boldsymbol{M}, \boldsymbol{\Lambda} + k\boldsymbol{I}) = p_B(\boldsymbol{q}; \boldsymbol{M}, \boldsymbol{\Lambda}), \forall k \in \mathbb{R}$.

*Proof.*

$$\begin{aligned}
p_B(\boldsymbol{q}; \boldsymbol{M}, \boldsymbol{\Lambda} + k\boldsymbol{I}) &= \frac{1}{Z(\boldsymbol{\Lambda} + k\boldsymbol{I})} \exp\left(\boldsymbol{q}^\top \boldsymbol{M}^\top (\boldsymbol{\Lambda} + k\boldsymbol{I}) \boldsymbol{M} \boldsymbol{q}\right) \\
&\propto \exp\left(\boldsymbol{q}^\top \boldsymbol{M}^\top (\boldsymbol{\Lambda} + k\boldsymbol{I}) \boldsymbol{M} \boldsymbol{q}\right) \\
&= \exp\left(\boldsymbol{q}^\top \boldsymbol{M}^\top \boldsymbol{\Lambda} \boldsymbol{M} \boldsymbol{q} + k\boldsymbol{q}^\top \boldsymbol{M}^\top \boldsymbol{M} \boldsymbol{q}\right) \\
&= e^k \exp\left(\boldsymbol{q}^\top \boldsymbol{M}^\top \boldsymbol{\Lambda} \boldsymbol{M} \boldsymbol{q}\right) \\
&\propto \exp\left(\boldsymbol{q}^\top \boldsymbol{M}^\top \boldsymbol{\Lambda} \boldsymbol{M} \boldsymbol{q}\right).
\end{aligned}$$

Given the normalization constant $Z(\boldsymbol{\Lambda})$, we have $p_B(\boldsymbol{q}; \boldsymbol{M}, \boldsymbol{\Lambda}+k\boldsymbol{I}) = \frac{1}{Z(\boldsymbol{\Lambda})} \exp\left(\boldsymbol{q}^\top \boldsymbol{M}^\top \boldsymbol{\Lambda} \boldsymbol{M} \boldsymbol{q}\right) = p_B(\boldsymbol{q}; \boldsymbol{M}, \boldsymbol{\Lambda})$. □

## C  Implementation Details for MOF-BFN

### C.1  Determining Local Geometries for Building Blocks

To determine the orientations of building blocks, we should first determine a reference frame for each block. A common-used solution is Principle Component Analysis (PCA) [4]. Given a building block $\mathcal{C}_j = (\boldsymbol{A}_j, \boldsymbol{X}_j)$, where $\boldsymbol{A}_j = [\boldsymbol{a}_r]_{r=1}^{N_j} \in \mathbb{R}^{h \times N_j}$, $\boldsymbol{X}_j = [\boldsymbol{x}_r]_{r=1}^{N_j} \in \mathbb{R}^{3 \times N_j}$ denote the atom types and coordinates within the block. The consistent local structure $\dot{\boldsymbol{X}}_j$ is defined as

$$\dot{\boldsymbol{X}}_j = \tilde{\boldsymbol{R}}\bar{\boldsymbol{X}}_j, \tag{28}$$

where $\bar{\boldsymbol{X}}_j = [\bar{\boldsymbol{x}}_r]_{r=1}^{N_j} = [\boldsymbol{x}_r - \frac{1}{N_j}\sum_{r=1}^{N_j}\boldsymbol{x}_r]_{r=1}^{N_j}$ is the centered coordinates, and $\tilde{\boldsymbol{R}} = \boldsymbol{c} \odot \boldsymbol{R}$. $\boldsymbol{R} = [\boldsymbol{e}_1, \boldsymbol{e}_2, \boldsymbol{e}_3]$ is the eigenvector matrix of $\bar{\boldsymbol{X}}_j\bar{\boldsymbol{X}}_j^\top$, and the signs $\boldsymbol{c} = [c_1, c_2, c_3]$ is determined by

$$c_i = \begin{cases} +1, \boldsymbol{e}_i^\top \boldsymbol{v}_j \geq 0, \\ -1, \boldsymbol{e}_i^\top \boldsymbol{v}_j < 0, \end{cases} \tag{29}$$

with a vector $\boldsymbol{v}_j$ determining the direction of the building block. Following [6], the vector is defined as $\boldsymbol{v}_j = \arg\min_{\bar{\boldsymbol{x}}_r, \|\bar{\boldsymbol{x}}_r\| \neq 0} \|\bar{\boldsymbol{x}}_r\|$.

### C.2  Rejection Sampling for Bingham Distribution

To sample from the Bingham distribution, we adopt a rejection sampling strategy using the Angular Central Gaussian (ACG) distribution as the proposal [5]. Given a multivariate normal distribution $\boldsymbol{u} \sim \mathcal{N}(0, \boldsymbol{\Sigma})$, the normalized direction $\boldsymbol{q} = \boldsymbol{q}/\|\boldsymbol{q}\|$ follows an ACG distribution, whose density is $p_{\mathrm{ACG}}(\boldsymbol{q}) \propto (\boldsymbol{q}^\top \boldsymbol{\Sigma}^{-1} \boldsymbol{q})^{-d/2}$, where $d$ is the dimension of the hypersphere. To perform rejection sampling, we choose $\boldsymbol{\Sigma}^{-1} = \boldsymbol{I} - 2\boldsymbol{\Lambda}$ such that the Bingham distribution is upper bounded by a scaled ACG distribution: $p_B(\boldsymbol{q}) \leq A' \cdot p_{\mathrm{ACG}}(\boldsymbol{q})$, with $A'$ related to the normalization term of each distribution. This bound leads to the condition

$$\exp(\boldsymbol{q}^\top \boldsymbol{\Lambda} \boldsymbol{q}) \leq A(\boldsymbol{q}^\top(\boldsymbol{I} - 2\boldsymbol{\Lambda})\boldsymbol{q})^{-d/2}$$
$$A \geq \frac{\exp(\boldsymbol{q}^\top \boldsymbol{\Lambda} \boldsymbol{q})}{(\boldsymbol{q}^\top(\boldsymbol{I} - 2\boldsymbol{\Lambda})\boldsymbol{q})^{-d/2}}$$
$$= \exp(\boldsymbol{q}^\top \boldsymbol{\Lambda} \boldsymbol{q})(1 - 2\boldsymbol{q}^\top \boldsymbol{\Lambda} \boldsymbol{q})^{d/2}$$

We define the function $f(t) = \exp(t)(1 - 2t)^{d/2}$ and maximize it over $t \in \mathbb{R}$ to obtain the tightest possible rejection bound. The maximum occurs at $t = \frac{1-d}{2}$, resulting in an optimal rejection constant $A_{\min} = \exp(\frac{1-d}{2})d^{d/2}$. Once a sample $\boldsymbol{q}$ is accepted, we apply a linear transformation $\boldsymbol{q}' = \boldsymbol{M}\boldsymbol{q}$ if the Bingham distribution has eigendecomposition $\boldsymbol{M}\boldsymbol{\Lambda}\boldsymbol{M}^\top$. This yields a sample from the desired Bingham distribution.

### C.3  Accuracy Scheduling for Bingham BFN

As the eigenvalue matrix $\boldsymbol{\Lambda}$ is diagonal, we denote the normalization term $Z(\boldsymbol{\Lambda})$ as $Z(\boldsymbol{\lambda})$ in this subsection. To determine a suitable value of $\alpha_i$, we consider the entropy of the Bingham distribution, which takes the form

$$H_i = H(\boldsymbol{\lambda}_i) = \log Z(\boldsymbol{\lambda}_i) - \boldsymbol{\lambda}_i^\top \nabla \log Z(\boldsymbol{\lambda}_i),$$

Assuming that $\boldsymbol{\lambda}_i \approx [0, -\beta_i, -\beta_i, -\beta_i]$, we can approximate the entropy as a function of a single parameter $\beta_i$. This assumption is an approximation that enforces isotropy around the principal axis, but it is effective in practice for constructing a simple and stable entropy scheduler. Let $\beta_0 = 0$ and $\beta_T$ sufficiently large, we linearly interpolate the entropy as $H_i = (1 - \frac{i}{T})H_0 + \frac{i}{T}H_T$, and numerically

solve for each intermediate value $\beta_i$. Consider the sender distribution $\boldsymbol{y}_i \sim p_W([1,0,0,0], \alpha_i)$, its second moment is given by

$$\mathbb{E}[\boldsymbol{y}_i \boldsymbol{y}_i^\top] = \text{diag}\Big( \frac{\nabla Z(0, -\alpha_i, -\alpha_i, -\alpha_i)}{Z(0, -\alpha_i, -\alpha_i, -\alpha_i)} + \big(1 - \sum \frac{\nabla Z(0, -\alpha_i, -\alpha_i, -\alpha_i)}{Z(0, -\alpha_i, -\alpha_i, -\alpha_i)}\big)[1,0,0,0]\Big).$$

From this, the expected change in $\beta$ is

$$\mathbb{E}[\beta_i - \beta_{i-1}] = \alpha_i(\mathbb{E}[\boldsymbol{y}_i \boldsymbol{y}_i^\top]_0 - \mathbb{E}[\boldsymbol{y}_i \boldsymbol{y}_i^\top]_1)$$
$$= \alpha_i\big(1 - 4\frac{\nabla Z(0, -\alpha_i, -\alpha_i, -\alpha_i)_1}{Z(0, -\alpha_i, -\alpha_i, -\alpha_i)}\big),$$

where the subscripts indicate the value at the corresponding indices. And we can numerically solve for $\alpha_i$ given $\beta_i$. Such scheduler gradually sharpens the Bingham distribution, increasing its concentration around the target direction, ensuring that the model starts with a high-entropy prior and becomes progressively confident as step increasing.

# D  Extension to De Novo Generation

## D.1  Implementation Details

The utilized MOF datasets contains millions of building block conformations, making a simple one-hot encoding for building blocks sparse and computationally inefficient. To obtain compact and continuous representations, a contrastive learning framework is employed in MOFDiff [3]. Each building block is encoded into a latent vector using a SE(3)-equivariant message passing neural network, specifically GemNet-OC.

Let $\mathcal{G}$ denote the set of all building blocks obtained from training MOFs. For each building block $\mathcal{C} \in \mathcal{G}$, the encoder network $f_\theta$ maps its local structure to a continuous embedding $\mathbf{z}_\mathcal{C} = f_\theta(\mathcal{C}) \in \mathbb{R}^d$, with latent dimension $d = 32$ in MOFDiff.

To ensure that the learned embeddings reflect chemical similarity, contrastive learning is performed using positive and negative pairs of building blocks. A positive pair $(\mathcal{C}, \mathcal{C}^+)$ consists of two blocks sharing the same ECFP4 fingerprint, while a negative pair $(\mathcal{C}, \mathcal{C}^-)$ indicates different block identities.

The contrastive learning objective is based on the following loss:

$$\mathcal{L}_{\text{contrast}} = -\sum_{i \in \mathcal{S}} \log \frac{\sum_{j \in \mathcal{S}+} \exp\left(\text{sim}(\mathbf{z}_i, \mathbf{z}_j)/\tau\right)}{\sum_{j \in \mathcal{S}} \exp\left(\text{sim}(\mathbf{z}_i, \mathbf{z}_j)/\tau\right)}, \tag{30}$$

where $\mathcal{S}, \mathcal{S}+$ denote the batch and the positive subset of the batch, $\tau > 0$ is a temperature hyperparameter, and $\text{sim}(\cdot, \cdot)$ denotes the cosine similarity:

$$\text{sim}(\mathbf{z}_1, \mathbf{z}_2) = \frac{\mathbf{z}_1^\top \mathbf{z}_2}{\|\mathbf{z}_1\| \|\mathbf{z}_2\|}, \tag{31}$$

While the continuous encoding space enables efficient retrieval via KD-Trees, we empirically observe that the distribution of embeddings deviates significantly from a normal distribution, which poses challenges for training a BFN. To address this, we normalize the representations as

$$\bar{z}_i = \frac{z_i - \text{mean}_{j \in \mathcal{G}}(z_j)}{\text{std}_{j \in \mathcal{G}}(z_j)}.$$

Similar to the BFN for lattice parameters, given the block embeddings $\bar{\boldsymbol{Z}} = [\bar{z}_i]_{i=1}^K$ of a structure, the input distribution is given by $\mathcal{N}(\bar{\boldsymbol{Z}}; \boldsymbol{\mu}_i^\mathcal{B}, (\rho_i^\mathcal{B})^{-1}\boldsymbol{I})$ parameterized by $\theta_i^\mathcal{B} = \{\boldsymbol{\mu}_i^\mathcal{B}, \rho_i^\mathcal{B}\}$. and the prior distribution is chosen as $\theta_0^\mathcal{B} = \{\mathbf{0}, 1\}$. After acquiring a sample from the sender distribution $\boldsymbol{y}_i^\mathcal{B} \sim \mathcal{N}(\bar{\boldsymbol{Z}}, (\alpha_i^\mathcal{B})^{-1}\boldsymbol{I})$ at step $i$, the Bayesian update function is

$$\{\boldsymbol{\mu}_i^\mathcal{B}, \rho_i^\mathcal{B}\} = \{\frac{\rho_{i-1}^\mathcal{B}\boldsymbol{\mu}_{i-1}^\mathcal{B} + \alpha_i^\mathcal{B}\boldsymbol{y}_i^\mathcal{B}}{\rho_{i-1}^\mathcal{B} + \alpha_i^\mathcal{B}}, \rho_{i-1}^\mathcal{B} + \alpha_i^\mathcal{B}\}. \tag{32}$$

The corresponding Bayesian flow distribution is accumulated as

$$p_F^{\mathcal{B}}(\boldsymbol{\mu}_i^{\mathcal{B}}|\bar{\boldsymbol{Z}}, i) = \mathcal{N}\big((1 - \sigma_T^{2i/T})\bar{\boldsymbol{Z}}, \sigma_T^{2i/T}(1 - \sigma_T^{2i/T})\boldsymbol{I}\big). \tag{33}$$

The training objective on the latent space is

$$\mathcal{L}_{\mathcal{B}} = \mathbb{E}_{i \sim U(1,T), \boldsymbol{\mu}_{i-1}^{\mathcal{B}} \sim p_F^{\mathcal{B}}(\boldsymbol{\mu}_{i-1}^{\mathcal{B}}|\bar{\boldsymbol{Z}}, i-1)} \Big[ \frac{\alpha_i^{\mathcal{B}} T}{2} \|\bar{\boldsymbol{Z}} - \phi_{\mathcal{B}}(\boldsymbol{\theta}_{i-1}^{\mathcal{M}}, i)\|_2^2 \Big]. \tag{34}$$

And the entire training objective is extended as

$$\mathcal{L}_{\text{DNG}} = \lambda_{\boldsymbol{\xi}} \mathcal{L}_{\boldsymbol{\xi}} + \lambda_{\boldsymbol{F}} \mathcal{L}_{\boldsymbol{F}} + \lambda_{\boldsymbol{q}} \mathcal{L}_{\boldsymbol{q}} + \lambda_{\mathcal{B}} \mathcal{L}_{\mathcal{B}}.$$

### D.2 Results on Relaxed Structures

Similar to MOFDiff, we further relax the generated structures via the UFF force field [7]. We refine both the lattice parameters and the all-atom positions by `LAMMPS` [9] and `LAMMPS Interface` [2]. The numbers of valid structures before and after relaxation are reported in Table 5, further demonstrating the generation quality of MOF-BFN.

Table 5: **Generation validity.** Number of structures that passed (↑) or failed (↓) each criterion among 1,000 generated candidates.

| Validity Criteria | Before Relaxation | | After Relaxation | |
|---|---|---|---|---|
| | MOF-BFN | MOFDiff | MOF-BFN | MOFDiff |
| *Connection Point Matching* | | | | |
| matched ↑ | **923** | 723 | **923** | 723 |
| *UFF Relaxation* | | | | |
| relaxed ↑ | - | - | **849** | 662 |
| *MOFChecker* | | | | |
| has_carbon ↑ | **923** | 723 | **849** | 662 |
| has_hydrogen ↑ | **900** | 715 | **827** | 654 |
| has_atomic_overlaps ↓ | **106** | 187 | **35** | 136 |
| has_overcoordinated_c ↓ | **178** | 264 | **8** | 17 |
| has_overcoordinated_n ↓ | 45 | **34** | **0** | **0** |
| has_overcoordinated_h ↓ | **166** | 247 | **20** | 21 |
| has_undercoordinated_c ↓ | **168** | 179 | **144** | 194 |
| has_undercoordinated_n ↓ | 142 | **91** | 141 | **133** |
| has_undercoordinated_rare_earth ↓ | **0** | **0** | **0** | **0** |
| has_metal ↑ | **923** | 723 | **849** | 662 |
| has_lone_molecule ↓ | **172** | 316 | **36** | 60 |
| has_high_charges ↓ | **64** | 104 | **5** | 13 |
| has_suspicious_terminal_oxo ↓ | **0** | 1 | **0** | 2 |
| has_undercoordinated_alkali_alkaline ↓ | 25 | **1** | **0** | **0** |
| has_geometrically_exposed_metal ↓ | 281 | **253** | **8** | 24 |
| *Total* | | | | |
| total_valid ↑ | **323** | 107 | **545** | 317 |

### D.3 Novelty and Uniqueness

Evaluating novelty and uniqueness is also critical in de novo generation. To address this, we additionally conduct an additional analysis using `MOFid`, a SMILES-style descriptor that captures both the building block identities and the topology of MOF structures. Specifically, we define novelty as the proportion of generated structures whose MOFid does not appear in the training set, and uniqueness as the proportion of generated structures with MOFids that are distinct from all other generated samples. To ensure that novelty is meaningful, we further define validity as structures that pass `LAMMPS` relaxation, satisfy all `MOFChecker` criteria, and are successfully processed by

MOFid. Table 6 summarizes the results. These results demonstrate that MOF-BFN attains a higher overall V.N.U. rate. Notably, MOFDiff exhibits a comparatively higher novelty and uniqueness ratio among valid samples, indicating a stronger tendency to generate structures distinct from the training distribution. However, this advantage is offset by a lower validity, as a substantial portion of the generated candidates fail to satisfy basic structural or physical constraints. It is therefore important to contextualize novelty within the scope of validity, since novel structures lacking physical plausibility are unlikely to be of practical significance for downstream applications. By contrast, MOF-BFN yields a substantially larger number of valid structures while preserving a considerable degree of novelty and uniqueness. Consequently, it achieves the highest proportion of valid, novel, and unique (V.N.U.) samples. This highlights a more balanced trade-off between structural diversity and physical realism, which is essential for practical applications in MOF design.

Table 6: **Generation VNU rates.** Validity, Novelty and Uniqueness of 1,000 generated samples based on MOFid.

| Model | Valid w/o MOFid | Valid w/ MOFid | Valid & Novel | Valid & Novel & Unique |
|---|---|---|---|---|
| MOFDiff | 323 | 284 | 281 | 281 |
| MOF-BFN | **545** | **450** | **424** | **407** |

### D.4 Geometric Property Evaluation

We further evaluate the generated samples in terms of the Wasserstein distances against the training set on geometric properties listed in Table 2. As shown in Table 7, the results indicate that the property distributions of MOF-BFN is closer to the training data compared to MOFDiff.

Table 7: **Geometric property evaluation.** Wasserstein distances computed between the training set and generated MOFs.

| | VSA ($m^2/cm^3$) | GSA ($m^2/g$) | AV ($Å^3$) | UCV ($Å^3$) | VF | PLD (Å) | LCD (Å) | DST ($g/cm^3$) |
|---|---|---|---|---|---|---|---|---|
| MOFDiff | 177.8 | 258.4 | 1039.7 | 4737.2 | 0.043 | **0.607** | 1.311 | 0.059 |
| MOF-BFN | **157.2** | **131.7** | **885.2** | **1626.3** | **0.022** | 0.665 | **1.177** | **0.043** |

## E   Extension to Conditional Generation

Enabling conditionl generation is also a valuable problem. To demonstrate this, we further conduct a preliminary experiment on void fraction (VF) via a classifier-guided generation (CFG)-style approach [8]. In this approach, the model was fine-tuned to accept an additional conditioning input $c$, representing the desired value of a target property. During generation, both an unconditional output $x$ and a conditional output $x_c$ were obtained. The final generated sample was then computed as $exp_x(w \log_x x_c)$, where $exp$ and $\log$ denote the exponential and logarithmic maps defined on the specific manifold, including the lattice representation, fractional coordinate space, unit quaternion space, and block embedding space. The weighting factor $w$ controls the degree of conditioning. Figure 4 shows that the void fraction distribution of the generated structures shifts significantly toward the desired value under conditioning.

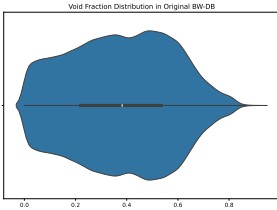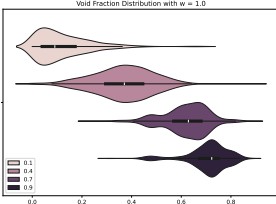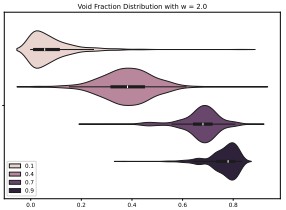

Figure 4: Void Fraction (VF) distribution of the original BW-DW dataset (left), the generated MOFs with $w = 1.0$ (middle) and $w = 2.0$ (right). The distributions shift significantly toward the desired value under conditioning, and larger guidance weights lead to stronger control.

## F  Additional Ablation Studies

**Orientation Representation**  In addition to unit quaternions, another widely used formulation for modeling 3D orientations is the axis–angle representation, which characterizes a rotation by a pair $(\boldsymbol{u}, \theta)$, where $\boldsymbol{u} \in \mathcal{S}^2$ denotes the rotation axis and $\theta \in [0, \pi)$ is the rotation angle. This representation is geometrically intuitive and has been extensively applied in rigid-body kinematics. To model this formulation within the proposed framework, we employ a decomposed strategy, where the rotation axis is represented by a von Mises–Fisher (vMF) distribution on the unit 2-sphere, while the rotation angle is represented by a von Mises distribution on the circle. This design preserves the intrinsic geometry of each component and enables independent Bayesian updates within the BFN framework.

Table 8: Additional ablation studies on orientation representations.

| Orientation Representation | Match Rate (%) | RMSD |
|---|---|---|
| Axis-Angle | 31.49 | 0.2770 |
| Unit Quaternion | 35.27 | 0.2735 |

The results on the structure prediction task are shown in Table 8. Despite being a theoretically valid representation, this alternative led to noticeably worse performance in structure prediction. As modeling axis and angle separately breaks the unified structure of the SO(3) manifold and complicates the joint learning of rotation distributions. In contrast, the quaternion representation allows a unified and conjugate-friendly update using the Bingham distribution. The additional results confirm that the use of unit quaternions and Bingham BFN is not only theoretically motivated but also empirically beneficial.

Table 9: Additional ablation studies on loss weights.

| $\gamma_q$ | $\gamma_{\mathcal{B}}$ | Valid before relaxation | Valid after relaxation |
|---|---|---|---|
| 0.2 | 10.0 | 323 | 54.5 |
| 1.0 | 10.0 | 219 | 40.9 |
| 0.2 | 1.0 | 233 | 44.4 |

**Loss weights**  For the DNG task, we further analyse the loss weights for the orientation $\gamma_q$ and the block embedding $\gamma_{\mathcal{B}}$. The results in Table 9 show that scaling either component improperly leads to a drop in validity.

## G  Experimental Details

Hyperparameters for the structure prediction (§ 5.1) and de novo generation (§ 5.3) are provided in Table 10. Baseline results in Table 1 and 2 are from MOFFlow [6], and the samples for calculating MOFDiff results in Table 3 are directly yielded from the official pre-trained checkpoint [4]. To ensure a fair comparison, we share the same split with MOFDiff, where 95% structures are used for training, and the remaining 5% are for validation. The structure prediction and de novo generation models are trained on 8 GPUs with 80 GB memories, and the training procedures take 136 and 152 GPU hours, respectively.

## H  Baseline Selection

We provide the original scope of different models in Table 11. For the structure prediction (SP) and property evaluation task, we follow the benchmark introduced in MOFFlow, and the selected baselines includes optimization-based methods (RS and EA), the full-atom model DiffCSP and the block-level model MOFFlow. Optimization-based methods are excluded from property evaluation as they fail to produce meaningful structures in Table 1. For de novo generation (DNG) task, it requires generating both block identities and their structure. Among existing methods, only MOFDiff supports this setting specifically for MOF, and is thus selected as the baseline.

## I  Limitations

While MOF-BFN presents a promising approach to hierarchical MOF structure prediction, several limitations remain that we leave to future work. First, our current framework treats each building block as a rigid body with a fixed local geometry. This rigid-body assumption simplifies the generative process but neglects the intrinsic conformational flexibility of many organic linkers and secondary

---

[4] https://github.com/microsoft/MOFDiff

Table 10: Hyperparameter settings for experiments.

| Building Block Encoder | | | | |
| --- | --- | --- | --- | --- |
| num_layers | node_dim | edge_dim | hidden_dim | max_radius |
| 4 | 64 | 64 | 64 | 5 |
| Coarse-Grained Structure Predictor | | | | |
| num_layers | hidden_dim | time_dim | num_freqs | |
| 6 | 512 | 128 | 64 | |
| BFN | | | | |
| $\beta_T^{\boldsymbol{\xi}}$ | $\beta_T^{\boldsymbol{F}}$ | $\beta_T^{\boldsymbol{q}}$ | $\beta_T^{\mathcal{B}}$ | $T$ |
| 1000 | 1000 | 200 | 1000 | 50 |
| $\gamma_{\boldsymbol{\xi}}$ | $\gamma_{\boldsymbol{F}}$ | $\gamma_{\boldsymbol{q}}$ | $\gamma_{\mathcal{B}}$ | |
| 1.0 | 1.0 | 0.2 | 10.0 | |
| Structure Prediction Training | | | | |
| lr | min_lr | plateau_factor | plateau_patience | Adam_betas |
| $5 \times 10^{-4}$ | $1 \times 10^{-4}$ | 0.6 | 30 | [0.9, 0.98] |
| epochs | batch_size | gradient_clip_val | weight_decay | |
| 1000 | 512 | 0.5 | 0.01 | |
| De Novo Generation Training | | | | |
| lr | min_lr | plateau_factor | plateau_patience | Adam_betas |
| $5 \times 10^{-4}$ | $1 \times 10^{-4}$ | 0.6 | 30 | [0.9, 0.98] |
| epochs | batch_size | gradient_clip_val | weight_decay | |
| 3000 | 512 | 0.5 | 0.01 | |

Table 11: **Model Scope.** SP and DNG denote structure prediction and de novo generation, respectively.

| Model | Atom-level SP | Block-level SP | Block-level DNG |
| --- | --- | --- | --- |
| RS & EA | ✓ | | |
| DiffCSP | ✓ | | |
| MOFFlow | | ✓ | |
| MOFDiff | | | ✓ |
| MOF-BFN | | ✓ | ✓ |

building units. For instance, MOFDiff [3] reports that over 2 million building block instances in the dataset correspond to only 242k unique molecular graphs, indicating that significant conformational diversity exists within each block type. Although our method supports the extension of building block vocabularies through a continuous embedding space, it does not yet account for conformation generation within each block. Integrating internal flexibility modeling into the current framework could further enhance the model. Second, our current work focuses on unconditional generation and structure prediction tasks, without explicitly incorporating guidance signals for specific downstream properties. In practice, many MOF design scenarios require property-oriented generation, such as optimizing for gas adsorption capacity or catalytic activity. However, techniques for guiding models toward desired property targets remain underexplored in the field of Bayesian Flow Networks. Developing conditional generation mechanisms within the BFN framework is an important direction for future research to enable targeted material discovery.

## J  Broader Impacts

This work contributes to the development of MOF structure prediction and design. It may benefit applications in gas storage, separation, and catalysis by enabling more efficient exploration of the chemical design space. By improving structure prediction accuracy and generation validity, it can potentially accelerate material discovery in a data-driven way. However, our model is trained and evaluated primarily on the BW-DB dataset [1], which may contain inherent biases in block types, structural motifs, or chemical compositions. As a result, the generalization ability of the model to underrepresented MOF types or application-specific domains could be limited. Care should be taken when applying the model beyond the scope of the training data.

## K  Code Availablility

Our codes are available at `https://github.com/jiaor17/MOF-BFN`.

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
