# OpenReview forum: "MOF-BFN: Metal-Organic Frameworks Structure Prediction via Bayesian Flow Networks"
_NeurIPS.cc/2025/Conference — NeurIPS 2025 poster_

### Official Review · Reviewer_7Smn · 2025-07-02

**Clarity:** 3
**Significance:** 3
**Originality:** 4
**Rating:** 4
**Confidence:** 3

**Summary:**

The authors of this paper address the challenging task of structure prediction for Metal-Organic Frameworks (MOFs). They highlight the intrinsic complexities of the problem and the limitations of existing models such as CDVAE, DiffCSP, MOFFlow, and CrysBFN. To overcome these challenges, they propose Bayesian Flow Networks (BFNs) and introduce MOF-BFN, a novel generative framework for hierarchical MOF structure prediction based on this approach.

**Questions:**

Check weaknesses

**Ethical Concerns:**

["NO or VERY MINOR ethics concerns only"]

**Limitations:**

Check weaknesses

**Quality:**

3

**Strengths And Weaknesses:**

**Strength:**
 - The paper is well-written and easy to follow, with a clearly motivated problem and sufficient background information to support the study.
- Overall methodology is novel. MOF-BFN operates in the fractional coordinate system to preserve periodicity. Also they models block orientations using unit quaternions sampled from Bingham BFNs—a novel generative module.
- The experimental setup of the paper is solid, Authors have experimented with 	state-of-the-art models on benchmark material generation tasks.
- Source code is available

**Weakness:**
- Many relevant works on Crystal generations are missing FlowLLM, TGDMat, SymmSD, DiffCSP++
- A recent ICLR work, TGDMat[1], has demonstrated promising performance on the crystal structure prediction (CSP) task using text guidance with a single sample in a diffusion model. Could you please evaluate your model's performance under similar text-guided settings, or alternatively, report results on the BW-DB dataset using TGDMat (Long) for a fair comparison?

[1] https://arxiv.org/abs/2503.00522

---

> ### Author Rebuttal · Authors · 2025-07-30
>
> We thank the reviewer for the innovative questions, and answer them as follows.
>
> > **W1: Many relevant works on Crystal generations are missing FlowLLM, TGDMat, SymmCD, DiffCSP++.**
>
> Thanks a lot for pointing out the missing related works! We agree that these works explore promising directions for inorganic crystal generation. Specifically, FlowLLM initializes flow matching process with a LLM-based prior. TGDMat innovatively introduces text conditions to the generative model. DiffCSP++ considers space group-based generation given specific Wyckoff position (WP) assignments, and SymmCD extends this by further enabling the generation of WPs. However, MOF generation is slightly distinct from the generation of inorganic materials, where the hierarchical modeling and the orientation determination are the key problems, which are not directly addressed by the aforementioned methods. We will provide more discussions on these works in the revised version.
>
> > **W2: A recent ICLR work, TGDMat, has demonstrated promising performance on the crystal structure prediction (CSP) task using text guidance with a single sample in a diffusion model. Could you please evaluate your model's performance under similar text-guided settings, or alternatively, report results on the BW-DB dataset using TGDMat (Long) for a fair comparison?**
>
> Thanks for the thoughtful suggestion. We agree that text-guided generation is an exciting direction, and TGDMat makes an important step toward integrating language models with crystal structure prediction. However, we would like to clarify that TGDMat is designed for full-atom generation of inorganic crystals, whereas our work focuses on hierarchical generation of MOFs, which presents very different challenges. In particular, results in Table 1 already include DiffCSP, a representative full-atom diffusion model. Its poor performance highlights the limitations of non-hierarchical methods when applied to complex systems like MOFs.
>
> Moreover, adapting text-guidance to MOF data faces practical challenges. TGDMat relies on RoboCrystallographer [A] to generate long descriptions. Applying this tool to MOFs is difficult in practice, as a single unit cell can contain hundreds of atoms, leading to long generation times (from tens of seconds to several minutes per sample) and resulting descriptions often exceed thousands of tokens. This makes it infeasible to apply to the full BW-DB dataset, which contains over 300,000 structures. In addition, current text representations do not fully capture the hierarchical nature of MOFs.
>
> While we believe text-guided MOF generation is a valuable and promising research direction, a systematic formulation of the description construction, benchmark design, and more reasonable evaluation metrics are required to fully tackle this problem. These are non-trivial and orthogonal to the main scope of this work, and we consider them important directions for future research.
>
> [A] Ganose, Alex M., and Anubhav Jain. "Robocrystallographer: automated crystal structure text descriptions and analysis." MRS Communications 9.3 (2019): 874-881.

---

> > ### Comment · Reviewer_7Smn · 2025-08-02
> >
> > Thank you for addressing all my comments. I suggest incorporating these observations regarding text-guided MOF generation in the "Potential Future Work" section and also citing the missing baseline papers. I have no further questions.

---

### Official Review · Reviewer_C95o · 2025-07-03

**Clarity:** 3
**Significance:** 3
**Originality:** 4
**Rating:** 5
**Confidence:** 4

**Summary:**

This paper introduces MOF-BFN, the first framework for MOF structure prediction that jointly models periodicity, position, and orientation using a Bayesian Flow Network (BFN). It improves prior work by:
1. Modeling periodicity explicitly using fractional coordinates
2. Enhancing performance by leveraging Bayesian flow instead of flow matching

It also presents a technical contribution by extending BFN to model rotations. Experiments demonstrate superiority over existing approaches in both structure prediction and de novo generation tasks.

**Questions:**

1. Have the authors considered comparing the training and inference costs of MOF-FracFlow and MOF-BFN under consistent settings (e.g., batch size, GPU type, total epochs)?
2. Could the authors give a more detailed breakdown of the model architecture? Also, how is the number of building blocks $K$ determined during de novo generation?
3. How does the model size of MOF-BFN compare to MOFFlow and MOFFrac-Flow (e.g., number of parameters)?
4. Line 210-211 mentions that $M_0$ is an arbitrary orthogonal matrix. Could the authors elaborate on this?
5. While the benefits of BFN are acknowledged, could the authors comment on the potential drawbacks of Bayesian flow (e.g., stability, scalability) compared to flow matching or diffusion? This question is asked purely out of curiosity.

**Ethical Concerns:**

["NO or VERY MINOR ethics concerns only"]

**Final Justification:**

The authors have addressed my concerns, so I am retaining my acceptance score. I did not raise it to 6 due to existing limitations, such as the rigid body assumption and lack of conditional generation, as noted in the Appendix.

**Limitations:**

Yes

**Quality:**

3

**Strengths And Weaknesses:**

# Strengths
- This is the first work to integrate SO(3) modeling within the BFN framework, offering a novel contribution by jointly modeling periodicity, lattice, and orientation for MOFs.
- The method supports unconditional generation and explicitly handles periodicity, both of which are not well-addressed in previous work.

# Weaknesses
- Insufficient architectural detail: Section 4.3's "Backbone Model for MOF-BFN" provides only a high-level overview and lacks a self-contained explanation of the model architecture. Moreover, the method for determining the number of building blocks $K$ for de novo generation is not explained in Appendix C.
- Potential computational inefficiency: While the method shows strong empirical performance, it likely incurs higher computational costs than flow-based methods due to the need for rejection sampling and posterior simulation over non-Euclidean spaces. A quantitative comparison of resource usage (e.g., training/inference time, memory, hardware) with baselines like MOF-FracFlow is necessary to assess trade-offs.

## Typos and minor issues
- Line 44: to jointly generates
- Line 102: Represtations -> Representations
- Footnote in page 3: represtation -> representation
- Line 162: Incomplete sentence: “Initialized from the prior $\mathbf{\theta}_0^{\mathcal{M}}$.”
- Line 175: Bayesain -> Bayesian

---

> ### Author Rebuttal · Authors · 2025-07-30
>
> We are grateful for the reviewer’s thoughtful comments and provide our responses below.
>
> > **W1: Insufficient architectural detail: Section 4.3's "Backbone Model for MOF-BFN" provides only a high-level overview and lacks a self-contained explanation of the model architecture. Moreover, the method for determining the number of building blocks K for de novo generation is not explained in Appendix C.**
> > **Q2: Could the authors give a more detailed breakdown of the model architecture? Also, how is the number of building blocks K determined during de novo generation?**
>
>
> We thank the reviewer for pointing this out. The backbone model contains two parts. The building block encoder is a strateforward implementation of EGNN without any changes. The coarse-grained structure predictor is adapted from CSPNet proposed in DiffCSP. Specifically, let $h\_i^{(s)}$ denote the node representation of node $i$ at layer $s$. The input encoding is given by $h\_i^{(0)}=\rho(\mathcal{E(C\_i)}, M, \kappa\_i, \lambda\_i, f\_{pos}(t))$, where $f\_{pos}$ is the time encoder and $\rho$ is an MLP to fuse all input features. The message passing process is followed by
> $$m\_{ij}^{(s)}=\varphi_m(h\_i^{(s-1)}, h\_j^{(s-1)}, \xi, \psi\_{FT}(f\_i-f\_j))$$
> $$h\_i^{(s)}=h\_i^{(s-1)}+ \varphi_h( h\_i^{(s-1)}, \sum_{j\in\mathcal{N}(i)}m\_{ij}^{(s)}),$$
> where $\varphi\_m, \varphi\_h$ are MLPs and $\psi\_{FT}$ is the Fourier embedding to capture periodicity. The final outputs are given by
>
> $$\hat{\xi}=\varphi\_{\xi}(\sum\_ih\_i^{(S)})$$
> $$\hat{f}\_i=f\_i+\varphi\_{f}(h\_i^{(S)})$$
> $$\hat{q}\_i=\text{Normalize}(\varphi_{q}(h\_i^{(S)})),$$
>
> where $\varphi_{\xi}, \varphi_{f}, \varphi_{q}$ are MLPs, $\hat{\xi}, \hat{f}_i, \hat{q}_i$ are the model output for each feature, and $S$ is the number of layers. We will provide more details in the appendix for better understanding.
>
> Regarding the choice of the number of building blocks K during de novo generation, we follow a standard approach used in molecule and crystal generative models, that is collecting the empirical distribution of K from the training set and sampling from it during generation. We will highlight this in the revised version.
>
> > **W2: Potential computational inefficiency: While the method shows strong empirical performance, it likely incurs higher computational costs than flow-based methods due to the need for rejection sampling and posterior simulation over non-Euclidean spaces. A quantitative comparison of resource usage (e.g., training/inference time, memory, hardware) with baselines like MOF-FracFlow is necessary to assess trade-offs.**
> > **Q1: Have the authors considered comparing the training and inference costs of MOF-FracFlow and MOF-BFN under consistent settings (e.g., batch size, GPU type, total epochs)?**
>
> Thank you for raising this question! To assess the additional computational cost introduced by BFN, we have additionally conducted a direct comparison between MOF-FracFlow and MOF-BFN under the same training and inference settings. Both models are trained on 8 GPUs (each with 80 GB memory) using a batch size of 512 for 1000 epochs. Inference is performed on a single GPU, with a batch size of 512 and 50 sampling steps. The GPU hours consumed during training and inference are summarized below:
>
> | Model | Training GPU hours | Inference GPU hours |
> |-|-|-|
> | MOF-FracFlow   |  135.7  | 0.22   |
> | MOF-BFN   | 137.5   | 0.50   |
>
> The additional cost of MOF-BFN arises primarily from posterior simulation on non-Euclidean manifolds. We further explore the efficiency of the rejection sampling detailed in Appendix B.2. Specifically, we sample 10,000 unit quaternions from a Bingham distribution with parameter matrix $\Lambda=-diag(0,\lambda, \lambda, \lambda)$ and measure the total GPU time as well as average number of rejection sampling iterations required per sample. The results are shown as follows.
>
> | $\lambda$ | GPU seconds | Avg. Iteration |
> |-|-|-|
> | 1 | 0.023 | 1.41 |
> | 10 | 0.028 | 1.90 |
> | 100 | 0.030 | 2.20 |
> | 1000 | 0.032 | 2.24 |
>
> Notably, rejection sampling is parallelizable on GPU and does not involve any model forward, making it acceptable for training and inference.
>
> > **Q3: How does the model size of MOF-BFN compare to MOFFlow and MOFFrac-Flow (e.g., number of parameters)?**
>
> All three models use the same EGNN-based block encoder, so differences in model size come primarily from the coarse-grained structure predictor. MOFFlow employs the OpenFold backbone, whereas MOF-FracFlow and MOF-BFN use the CSPNet backbone, which is coupled with the selections of the coordinate system. The number of parameters are detailed as follows:
>
> | Model | # of params (M) |
> |-|-|
> | MOFFlow | 22.5 |
> | MOF-FracFlow | 11.2 |
> | MOF-BFN | 11.2 |
>
> This shows that MOF-BFN achieves better performance with fewer parameters compared to MOFFlow.
>
> > **Q4: Line 210-211 mentions that $M_0$ is an arbitrary orthogonal matrix. Could the authors elaborate on this?**
>
> Thanks for the question. Theoretically, when the Bingham concentration matrix is initialized as $\Lambda=0$, the matrix $A_0=M_0^\top\Lambda M$ becomes a zero matrix regardless of the choice of orthogonal matrix $M_0$, resulting in a uniform distribution over the unit sphere. In practice, we initialize $M_0$ by performing `torch.linalg.eigh` on the zero matrix, which yields the identity matrix $M_0=I$. We will clarify this implementation detail in the revised version to avoid confusion.
>
> > **Q5: While the benefits of BFN are acknowledged, could the authors comment on the potential drawbacks of Bayesian flow (e.g., stability, scalability) compared to flow matching or diffusion? This question is asked purely out of curiosity.**
>
> Thank you for this insightful question. While Bayesian Flow Networks offer strong performance, we acknowledge that there are still open challenges and potential limitations compared to alternative generative models. One key drawback is that Bayesian updates must be explicitly designed for each target distribution, which can require nontrivial mathematical derivations, especially on non-Euclidean manifolds. There is currently no general framework for constructing BFN updates on arbitrary manifolds. We believe this is an exciting research direction for the future to broader the generalizability of BFNs.
>
> > **Typos and Miner Issues**
>
> Thanks for pointing these out. We will correct these in the revised version.

---

> ### Comment · Reviewer_C95o · 2025-08-05
>
> Thank you for your detailed response to my comments and questions. My concerns have been addressed. I retain my score for acceptance.

---

### Official Review · Reviewer_Cxvg · 2025-07-03

**Clarity:** 3
**Significance:** 3
**Originality:** 4
**Rating:** 5
**Confidence:** 4

**Summary:**

The paper introduces MOF-BFN, a generative model for predicting the structures of Metal-Organic Frameworks (MOFs) using Bayesian Flow Networks (BFNs). MOF-BFN leverages the local geometry of metal nodes and organic linkers to jointly predict lattice parameters, as well as the positions (in fractional coordinates) and orientations (via unit quaternions) of building blocks within a periodic unit cell. The model uses a Bingham BFN to effectively sample orientations on the 4D unit hypersphere. Experiments show MOF-BFN achieves strong performance in structure prediction, geometric property evaluation, and de novo MOF generation.

**Questions:**

1. How diverse are the generated MOF structures? Can you provide an analysis showing whether the generated topologies differ significantly from those in the training dataset?
2. Have you evaluated whether the distribution of geometric and physical properties in the generated structures shifts relative to the training set? If so, can you discuss the implications of any observed shifts?
3. Could you elaborate on the novelty aspect of your de novo generation results? How do you measure or ensure that newly generated structures represent novel topologies rather than variants of known ones?
4. What are the main limitations of your approach, particularly regarding generation validity rates and scalability? How might these limitations affect the applicability of MOF-BFN in real-world MOF design tasks?

**Ethical Concerns:**

["NO or VERY MINOR ethics concerns only"]

**Final Justification:**

The authors have addressed all my concerns in the rebuttal

**Limitations:**

Section Missing

**Quality:**

3

**Strengths And Weaknesses:**

Strengths:

1. The paper tackles the complex task of MOF structure generation using Bayesian Flow Networks, extending the paradigm to a challenging problem.
2. The paper has a sound modeling strategy and architecture design; the use of fractional coordinates and BFNs provides a novel alternative to conventional ODE and SDE-based models. The model appropriately incorporates relevant symmetries, improving its ability to handle the periodic nature of MOFs.
3. The model is compared against relevant baselines and demonstrates strong performance across the reported metrics, highlighting its effectiveness relative to existing approaches.

Weaknesses:

1. The paper does not discuss the diversity or uniqueness of the generated MOF structures. It remains unclear whether the model generates topologies that are distinct from those in the training set.
2. There is no analysis of whether the generated structures deviate in terms of geometric or physical properties (e.g., as shown in Table 2) from the training data. A meaningful distribution shift is often a desirable characteristic in generative models, as it indicates the ability to explore novel and diverse regions of the design space.
3. The de novo generation results focus on structural validity rather than novelty. However, novelty is a central goal of de novo design and should be explicitly evaluated and discussed.
4. The authors do not adequately address the limitations of their method. Given the complexity of MOF generation and the relatively modest validation rates, it is important to acknowledge the method’s current boundaries and potential areas for improvement.

---

> ### Author Rebuttal · Authors · 2025-07-30
>
> We thank the reviewer for the insightful comments, and address the concerns as follows.
>
> > **W1: The paper does not discuss the diversity or uniqueness of the generated MOF structures. It remains unclear whether the model generates topologies that are distinct from those in the training set.**
> > **W3: The de novo generation results focus on structural validity rather than novelty. However, novelty is a central goal of de novo design and should be explicitly evaluated and discussed.**
> > **Q1: How diverse are the generated MOF structures? Can you provide an analysis showing whether the generated topologies differ significantly from those in the training dataset?**
> > **Q2: Could you elaborate on the novelty aspect of your de novo generation results? How do you measure or ensure that newly generated structures represent novel topologies rather than variants of known ones?**
>
> We thank the reviewers for raising the important question of novelty and uniqueness in de novo generation. To address this, we have futher conducted an additional analysis using MOFid [A], a SMILES-style descriptor that captures both the building block identities and the topology of MOF structures.
>
> We define novelty as the proportion of generated structures whose MOFid does not appear in the training set, and uniqueness as the proportion of generated structures with MOFids that are distinct from all other generated samples. To ensure that novelty is meaningful, we further define validity as structures that pass LAMMPS relaxation, satisfy all MOFChecker criteria, and are successfully processed by MOFid. The following table summarizes the results.
>
>
> | Model | Valid | Valid & Novel | Valid & Novel & Unique |
> |-|:-:|:-:|:-:|
> | MOFDiff   | 28.4 %  | 28.1 %   |  28.1 % |
> | MOF-BFN   | 45.0 %   | 42.4 %   | 40.7 % |
>
> These results show that MOF-BFN achieves a higher overall V.N.U. rate. It is worth noting that MOFDiff shows a relatively higher novelty and uniqueness ratio among valid samples. This suggests that it tends to generate structures that differ more from the training data. However, this comes with a lower overall validity, which means many generated samples do not meet basic structural or physical constraints. We believe that novelty should be considered in the context of validity, as novel structures that are not physically plausible may not be meaningful for downstream applications. In contrast, MOF-BFN achieves a significantly higher number of valid structures while maintaining a strong level of novelty and uniqueness. As a result, it achieves the highest proportion of valid, novel, and unique (V.N.U.) samples. We believe this reflects a more balanced trade-off between generating diverse structures and ensuring structural realism, which is essential for practical MOF design.
>
> > **W2: There is no analysis of whether the generated structures deviate in terms of geometric or physical properties (e.g., as shown in Table 2) from the training data. A meaningful distribution shift is often a desirable characteristic in generative models, as it indicates the ability to explore novel and diverse regions of the design space.**
> > **Q2: Have you evaluated whether the distribution of geometric and physical properties in the generated structures shifts relative to the training set? If so, can you discuss the implications of any observed shifts?**
>
> We thank the reviewers for highlighting the importance of analyzing distributional shifts in the generated structures. We further evaluate the generated samples in terms of the Wasserstein distances against the training set on geometric properties listed in Table 2. As shown in the following table, the results indicate that the property distributions of MOF-BFN is closer to the training data compared to MOFDiff.
>
> | Model | PLD | LCD | GSA | VSA | UV | DST | AV | VF |
> |-|-|-|-|-|-|-|-|-|
> | MOFDiff   | 0.607 | 1.311 | 258.4 | 177.8 | 4737.2 | 0.059 | 1039.7 | 0.043
> | MOF-BFN   | 0.665 | 1.177 | 131.7 | 157.2 | 1626.3 | 0.043 | 885.2 | 0.022 |
>
> Moreover, we agree that enabling controlled deviation from the training distribution is also valuable. To demonstrate this, we conduct a preliminary experiment on void fraction (VF), where we implement a classifier-guided generation (CFG)-style approach [B]. Specifically, we fine-tune the model to accept an additional conditioning input $c$, where $c$ is the value of the desired value of VF. At generation time, we combine the unconditional output $x$ and the conditional output $x_c$ into $\exp_x (w\log_x x_c)$, where $\exp$ and $\log$ denote the exponential and logarithmic maps on the specific manifold (lattice representation, fractional coordinates, unit quaternions, and block embedding space), and $w$ is a weighting factor. We evaluate the guided samples based on their median VF and the Wasserstein distance from the training distribution.
>
>
> | w | Conditioned VF Value | Median | W. Dist |
> | :-: | :-: | :-: | :-: |
> | 1.0 | 0.1 | 0.091 | 0.258 |
> | | 0.4 | 0.371 | 0.076 |
> | | 0.7 | 0.629 | 0.241 |
> | 2.0 | 0.1 | 0.055 | 0.298 |
> | | 0.4 | 0.382 | 0.089 |
> | | 0.7 | 0.681 | 0.292 |
>
> As shown above, the void fraction distribution of the generated structures shifts significantly toward the desired value under conditioning. It is also worth noting that all conditioned samples have larger Wasserstein distances than the unconditional model (0.022). and the distance grows with larger guidance weight $w$. These reflect a natural trade-off that, as we enforce stronger property control, the generated structures deviate further from the original data distribution. This suggests that MOF-BFN can not only reproduce training distributions but also be guided to explore specific property-conditioned structures.
>
> > **W4: The authors do not adequately address the limitations of their method. Given the complexity of MOF generation and the relatively modest validation rates, it is important to acknowledge the method’s current boundaries and potential areas for improvement.**
> > **Q4: What are the main limitations of your approach, particularly regarding generation validity rates and scalability? How might these limitations affect the applicability of MOF-BFN in real-world MOF design tasks?**
>
> We appreciate the reviewer’s question. We have already provided a dedicated discussion in Appendix E, which outlines current limitations and future directions on two key aspects. First, our current framework models each MOF building block as a rigid body. This assumption simplifies the generative process and enables more stable orientation modeling, but it may neglects the intrinsic conformational flexibility of the MOF structures. Future works could extend the model to allow flexible generation of building blocks. Second, while MOF-BFN achieves strong unconditional generation performance, it does not yet fully explore property-guided generation, which is highly relevant for real-world inverse design. As a first step, we have conducted a pilot study as above (response to W2/Q2). This shows promising results, but a more systematic framework for property-guided generation remains an important avenue for future development.
>
> [A] Bucior, Benjamin J., et al. "Identification schemes for metal–organic frameworks to enable rapid search and cheminformatics analysis." Crystal Growth & Design 19.11 (2019): 6682-6697.
>
> [B] Tao, Nianze, and Minori Abe. "Bayesian flow network framework for chemistry tasks." Journal of Chemical Information and Modeling 65.3 (2025): 1178-1187.

---

> > ### Comment · Reviewer_Cxvg · 2025-08-05
> >
> > The authors have adequately addressed my concerns, and I have no further comments. I have therefore raised my score.

---

### Official Review · Reviewer_hsw3 · 2025-07-03

**Clarity:** 4
**Significance:** 3
**Originality:** 3
**Rating:** 5
**Confidence:** 2

**Summary:**

The authors introduce a novel way of parameterizing a joint lattice-position-orientiation distribution of hierarchical MOF structures. By simultaneously learning the bayesian flow distributions of the lattice parameters in R^6,  the fractional coordinates represented by elements of the torus T^3 (times the number of bodies), and the bingham distribution on the unit quaternions (again times the number of bodies), the authors demonstrate the ability to generate better structures compared to several baselines.

**Questions:**

-Do there exist families of distributions on other Lie groups such as $SE(3)$? The nice thing about using the unit quaternions, which are specific to SO(3), is that the Bingham distribution defined on them has a nice conjugate relationship with the Watson distribution.

-Could the role of $\mathcal{L}_{\mathcal{B}}$ be clarified for doing de novo generation?

-How does the block representation of MOFFlow compare to the representation proposed in the paper? If they are different, which (between the block representation being used or the learning algorithm itself) is the dominant factor in model performance?

-In general, what else is different between MOF-BFN and MOFFlow, and why would we expect that change to make MOF-BFN superior?

**Ethical Concerns:**

["NO or VERY MINOR ethics concerns only"]

**Final Justification:**

I had some questions about the original paper which the authors answered clearly and thoroughly, so I have decided to increase my score as a result.

**Limitations:**

Yes

**Paper Formatting Concerns:**

Good

**Quality:**

3

**Strengths And Weaknesses:**

Strengths:
-The paper is straightforward to read and understand, and offers a clearly novel idea for learning to sample hierarchical MOF structures. The results demonstrate clear improvement over existing approaches.

-The authors use a clever conjugate relationship to enable bayesian flow distribution to be defined on the unit quaternions

Weaknesses:
-One thing i don’t fully understand is how widespread of a tool this would be for MOF researchers. I don’t fully understand the scope of problems this model can solve. Are there examples of this?

-It seems like it would be eventually be useful to have some kind of MOF simulator (e.g. DFT) in the loop to provide effectively infinite data for the model to learn from.

-Not sure how convincing the margin between MOFFlow and MOFBFN is.

---

> ### Author Rebuttal · Authors · 2025-07-30
>
> We thank the reviewer for the valuable comments, and answer the questions as follows.
>
> > **W1: One thing I don’t fully understand is how widespread of a tool this would be for MOF researchers. I don’t fully understand the scope of problems this model can solve. Are there examples of this?**
>
> We thank the reviewer for raising this important question. The proposed framework supports both structure prediction and de novo generation for MOFs, and enables several practical applications. Firstly, MOF-BFN can generate diverse candidates can be used to populate virtual databases for high-throughput screening pipelines. Secondly, as a generative model, MOF-BFN can serve as a structure initialization tool that can complement or accelerate DFT-based optimization pipelines. Last, by generating novel combinations of nodes and linkers, our model can help researchers identify unconventional MOFs with potentially superior performance, especially for applications like gas storage, separation, or catalysis. Overall, MOF-BFN provides a powerful and versatile generative framework for MOF design, we believe this makes it a valuable tool for the MOF community.
>
> > **W2: It seems like it would be eventually be useful to have some kind of MOF simulator (e.g. DFT) in the loop to provide effectively infinite data for the model to learn from.**
>
> We agree that incorporating physics-based simulators such as DFT into the training loop is a promising future direction, but it is impractical to run them on the fly given the limited computational resouces. Hence, our current focus is on learning from existing structural data in a data-driven, simulator-free setting, in order to prove that our model has the capability to capture meaningful physical knowledge from the structural data. Looking ahead, we agree that simulators could be integrated into MOF-BFN in several post-training ways, such as fine-tuning the model on simulated properties, or incorporating reinforcement learning with DFT-based rewards. We consider this a highly promising extension and plan to explore it in future work.
>
> > **W3: Not sure how convincing the margin between MOFFlow and MOFBFN is.**
>
> Thanks for raising this concern. To better compare MOFFlow and MOF-BFN, we further report the mean, standard deviation, and the full range (min and max) of each metric across the five sampling runs used in Table 1. As shown below, MOF-BFN consistently outperforms MOFFlow in both match rate and RMSD across all runs, and the performance margin is statistically stable:
>
> |  |  | Match Rate (%) | RMSD |
> | :-: | :-: | :-: | :-: |
> | MOFFlow | Reported | 31.69 | 0.2820 |
> | MOF-BFN | Mean | 35.26 | 0.2751 |
> | | Std. | 0.10 | 0.0012 |
> | | Min. | 35.08 | 0.2735 |
> | | Max. | 35.38 | 0.2772 |
>
> > **Do there exist families of distributions on other Lie groups such as SE(3)?**
>
> We appreciate the reviewer’s insightful question. In our work, we decompose the state of each building block into its coordinate and orientation, corresponding to a periodic vector in $\mathcal{T}^3$ and a rotation in SO(3), respectively. This separation allows us to adopt conjugate-friendly distribution families that are well-suited for each component. Directly modeling probability distributions on SE(3) as a whole, and determining the Bayesian updates on it, remains an open research problem. Our current decomposition-based approach offers a practical alternative, and more unified treatments over SE(3) could be an interesting direction for future work.
>
> > **Could the role of $\mathcal{L_B}$ be clarified for doing de novo generation?**
>
> We thank the reviewer for the question. The role of $\mathcal{L_B}$ in de novo generation is already discussed in Appendix C.2. Specifically, $\mathcal{L_B}$ is the training objective of a continuous-space BFN, applied on the continuous latent representation learned by contrastive learning. This objectve enable the model to generate building blocks in the embedding space.
>
> > **How does the block representation of MOFFlow compare to the representation proposed in the paper? If they are different, which (between the block representation being used or the learning algorithm itself) is the dominant factor in model performance?**
>
> Thanks for pointing this out. Our model uses exactly the same block representation module as MOFFlow, which is a 4-layer EGNN. This design allows us to fairly evaluate the impact of our proposed generative approach.
>
> > **In general, what else is different between MOF-BFN and MOFFlow, and why would we expect that change to make MOF-BFN superior?**
>
> We appreciate the reviewer’s question. As noted earlier, both models share the exact same block representation model, ensuring a fair comparison. The key differences lie in the generative modeling framework. MOF-BFN introduces two major improvements over MOFFlow:
>
> - **Fractional coordinate modeling:** MOFFlow models building block positions in Cartesian coordinates, and MOF-BFN replaces this with a fractional coordinate system, enabling more accurate modeling of periodicity.
> - **Using of Bayesian Flow Network:** Instead of flow matching, MOF-BFN adopts the BFN framework, which defines explicit Bayesian updates over intermediate distributions. This leads to better generation performance.
>
> To demonstrate the contribution of each modification, we provided a controlled ablation study in Section 5.4. We compared:
>
> - MOFFlow (Baseline)
> - MOF-FracFlow (Fractional coordinates + Flow Matching)
> - MOF-BFN (Fractional coordinates + BFN)
>
> The results show a clear and consistent improvement at each stage.

---

> > ### Comment · Reviewer_hsw3 · 2025-08-03
> >
> > Thank you to the authors for the detailed answers. The improvement of MOF-BFN over MOFFlow is much clearer now, and thus I have decided to raise my score to a 5, as this is a technically sound paper with solid evidence to back it up.

---

### Official Review · Reviewer_7ek4 · 2025-07-06

**Clarity:** 3
**Significance:** 3
**Originality:** 2
**Rating:** 5
**Confidence:** 3

**Summary:**

This paper addresses Metal Organic Framework structure prediction and de novo generation. To tackle this, it leverages Bayesian Flow Networks that operate on a carefully selected MOF representation space, crucially, this space entails fractional coordinates, as well as orientations modeled as unit quaternions. These quaternions are parameterized via Bingham distributions.

**Questions:**

- Could the authors elaborate on the choice of unit quaternions and the Bingham distribution? It would be helpful to see empirical evidence, such as additional ablation studies, to justify the contributions.

- I am wondering if all four distinct hyperparameters are necessary to balance the loss function. Typically, such balancing is used when optimizing disparate training signals, whereas here each loss component appears to correspond to a core aspect of the representation.

**Ethical Concerns:**

["NO or VERY MINOR ethics concerns only"]

**Final Justification:**

The authors were able to answer my questions and provided convincing ablations during the rebuttal. Hence, I recommend acceptance for this paper.

**Limitations:**

Yes.

**Paper Formatting Concerns:**

- The relevant appendix sections should be cited across the main paper
- The hyperparameters in Eq.26 should not use lambda but preferably another letter. As this can lead to confusion with the lambda used for the eigenvalues represented before
- “At each step $i$ , the receiver treats its distribution from the previous step as the input” and Eq. 2 display notation that is not consistent with the original BFN work, as the input distribution is not used during sampling. Effectively, what gets updated is $p_R$ and not $p_I$. I find this confusing and would preder if the notation was in line with the original work.
- It should be clear just by looking at Figure 3 that MOF-FracFlow is an ablation
- In the related work or appendix, there should be a table that compares all external models and MOF-BFN with respect to their capabilities. This should be clear in the experimental section, since some of the tables/figures only display a subset of the comparable models

**Quality:**

3

**Strengths And Weaknesses:**

### Strengths:
- [Quality] The authors provide a theoretically sound approach to the representation of MOFs, which is further complemented by the selection of parameterization that contains key inductive biases connected to the application domain. The experimental results support the use of Bayesian Flow Networks, the proposed representations, and key parameterized distributions.
- [Clarity] In general, the submission is well-structured, with precise and consistent notation that highlights the key design choices and theoretical motivations. The experimental setup is easy to follow, and the results are well-presented.
- [Significance] MOF-BFN outperforms previous work, with selected representations that provide valuable insights for future work. Furthermore, it provides an overview of how to implement BFNs with the Bingham distribution. Therefore, it constitutes a meaningful achievement within the field
- [Originality[ The authors cite relevant work, and their contributions are well-contextualized within the field of MOFs and broad material design. The contributions are novel.
### Weaknesses:
- [Quality] It is scientifically unclear which of the technical contributions leads to improved experimental results.
- [Clarity] The large number of mathematical symbols makes reading and comprehension harder; Additionally, in the experimental section, it should be clear why only a subset of the external models are considered for each table/figure, please see my further remarks.
- [Significance] I believe the paper would benefit from more extensive ablation studies. In particular, with respect to the selection of unit quaternions and the utilization of the Bingham distribution, instead of alternative representations and distributions

---

> ### Author Rebuttal · Authors · 2025-07-30
>
> We thank the reviewer for the valuable comments, and we answer the reviewer's questions as follows.
>
> > **W1: It is scientifically unclear which of the technical contributions leads to improved experimental results.**
>
> Thank you for pointing this out. In the original submission, we provided an ablation study in Sec. 5.4 to isolate the contributions of the two key components. In particular, we compare MOFFlow, MOF-FracFlow, which incorporates fractional coordinates into the flow matching framework, and our MOF-BFN, which additionally uses Bayesian Flow Networks and quaternion orientation modeling. This progression helps attribute the performance gain. Using fractional coordinates leads to better handling of periodicity, and switching from flow matching to BFNs brings further gains in generative performance.
>
> In addition, we conducted a further ablation study on orientation representation by **replacing unit quaternions with axis-angle representations**. This allows us to assess the importance of using a principled and symmetric representation for modeling orientations. The details of this comparison are provided in our later reply to W3.
>
> > **W2.1: The large number of mathematical symbols makes reading and comprehension harder**
>
> We appreciate the reviewer’s suggestion and fully acknowledge the trade-off between technical precision and readability. To improve clarity, we will include a summary table in the appendix in the revised version to clarify the meaning of each key notation used in the equations.
>
> > **W2.2: In the experimental section, it should be clear why only a subset of the external models are considered for each table/figure.**
> > **In the related work or appendix, there should be a table that compares all external models and MOF-BFN with respect to their capabilities. This should be clear in the experimental section, since some of the tables/figures only display a subset of the comparable models.**
>
> We thank this insightful advice. Regarding the baselines, each subset is selected based on the applicability of the corresponding method to the specific task.
> - Tables 1 and 2 follow the benchmarks introduced in MOFFlow [A], which focuses on evaluating **MOF structure prediction** in terms of both structural accuracy and property alignment. The selected baselines includes optimization-based methods (RS and EA), the full-atom model DiffCSP and the block-level model MOFFlow. Optimization-based methods are excluded from property evaluation (Table 2) as they fail to produce meaningful structures in Table 1.
> - Table 3 evaluates **de novo generation**, which requires generating both block identities and their structure. Among existing methods, only MOFDiff supports this setting, and is thus selected as the baseline.
>
> We agree that a capability comparison table would help clarify the scope of each baseline, and we will add such a table in the revised version.
>
> > **W3: I believe the paper would benefit from more extensive ablation studies. In particular, with respect to the selection of unit quaternions and the utilization of the Bingham distribution, instead of alternative representations and distributions**
> > **Q1: Could the authors elaborate on the choice of unit quaternions and the Bingham distribution? It would be helpful to see empirical evidence, such as additional ablation studies, to justify the contributions.**
>
> As suggested by the reviewer, we explored alternative representations for modeling 3D orientations. In addition to unit quaternions, one common formulation is **the axis-angle representation**, which describes a rotation as a pair $(u, \theta)$, where $u\in\mathcal{S}^2$ is the rotation axis and $\theta\in[0,\pi]$ is the rotation angle. This representation is geometrically intuitive and widely used in rigid-body kinematics. To model this representation, we adopted a decomposed approach, where the rotation axis was modeled by a von Mises–Fisher (vMF) distribution on the 2-sphere, and the rotation angle by a von Mises distribution on the circle. This design preserves the geometry of each component and enables separate Bayesian updates within the BFN framework. The results on structure prediction task are shown as follows:
>
> | Orientation Representation | Match Rate (%) | RMSD |
> | :-: | :-: | :-: |
> | Axis-Angle | 31.49 | 0.2770 |
> | Unit Quaternion | 35.27 | 0.2735 |
>
> Despite being a theoretically valid representation, this alternative led to noticeably worse performance in structure prediction. As modeling axis and angle separately breaks the unified structure of the SO(3) manifold and complicates the joint learning of rotation distributions. In contrast, the quaternion representation allows a unified and conjugate-friendly update using the Bingham distribution. The additional results confirm that our use of unit quaternions and Bingham BFN is not only theoretically motivated but also empirically beneficial. We will include them in the revised version.
>
> Moreover, we also considered directly modeling rotation matrices. However, defining efficient and closed-form Bayesian updates over SO(3) matrices remains underexplored in the context of generative modeling. We believe this is a promising direction for future research.
>
> > **I am wondering if all four distinct hyperparameters are necessary to balance the loss function. Typically, such balancing is used when optimizing disparate training signals, whereas here each loss component appears to correspond to a core aspect of the representation.**
>
> Thanks for the insightful question. Although we did not perform extensive hyperparameter tuning, we analyse the weights for the orientation $\gamma_q$ and the block embedding $\gamma_\mathcal{B}$. The results below show that scaling either component improperly leads to a drop in validity.
>
> | $\gamma_q$ | $\gamma_\mathcal{B}$ | Validity before relaxation | Validity after relaxation |
> | :-: | :-: | :-: | :-: |
> | 0.2 | 10.0 | 32.3 % | 54.5 % |
> | 1.0 | 10.0 | 21.9 % | 40.9 % |
> | 0.2 | 1.0 | 23.3 % | 44.4 % |
>
>
>
>
> > **The relevant appendix sections should be cited across the main paper.**
>
> We thank the reviewer's suggestion and will refer the appendices in the main paper.
>
> > **The hyperparameters in Eq.26 should not use lambda but preferably another letter. As this can lead to confusion with the lambda used for the eigenvalues represented before.**
>
> Nice suggestion! We will replace the loss weight notations into $\gamma$ to avoid misleading.
>
> > **“At each step $i$, the receiver treats its distribution from the previous step as the input” and Eq. 2 display notation that is not consistent with the original BFN work, as the input distribution is not used during sampling. Effectively, what gets updated is $p_R$ and not $p_S$. I find this confusing and would prefer if the notation was in line with the original work.**
>
> Thanks for pointing this out. Our Eq. (2) was intended only as an intuitive illustration of the Bayesian update process. Indeed, the input distribution is not directly sampled during generation, but instead iteratively refined through parameter updates. Besides, as we stated in lines 145–149, since the sender distribution is unavailable during sampling, the receiver $p_R$ rather than $p_S$ is used. We will revise our notation and explanations to make this distinction clearer.
>
> > **It should be clear just by looking at Figure 3 that MOF-FracFlow is an ablation.**
>
> Thanks! We will clarify this in the caption of Figure 3 to make the readers directly understand the figure.

---

> > ### Comment · Reviewer_7ek4 · 2025-08-05
> >
> > I thank the authors for their efforts in answering my questions and concerns.
> >
> > - R1-R3: The authors' new ablation study is a great addition that really clarifies the key contributions of the paper. Their analysis justifies the use of unit quaternions, and I share their excitement about the closed-form Bayesian updates for SO(3) matrices.
> > - R4: The authors provide anecdotal evidence towards the hyperparameter selection. Although not rigorous, it provides a good empirical justification given the time constraints of the review process.
> > - R2.1-R2.2, R5-R8: I believe the suggested changes for these sections will definitely improve the paper's clarity.
> >
> > I have no further concerns and recommend acceptance for this paper.

---

### Author Response · Authors · 2025-08-09
**General Response**

Dear ACs and reviewers,

Thanks for your time and efforts in reviewing and discussing this paper. This paper mainly focuses on the structure prediction and de novo generation of Metal-Organic Frameworks (MOFs) via Bayesian Flow Networks (BFNs). We are glad that the reviewers recognized the contributions of our paper, which we briefly summarize as follows.

- **Novelty:** "The contributions are novel." (7ek4) "The use of fractional coordinates and BFNs provides a novel alternative to conventional ODE and SDE-based models." (Cxvg) "This is the first work to integrate SO(3) modeling within the BFN framework." (C95o)
- **Presentation:** "In general, the submission is well-structured." (7ek4) "The paper is straightforward to read and understand." (hsw3) "The paper is well-written and easy to follow." (7smn)
- **Experiments:** "The model is compared against relevant baselines and demonstrates strong performance across the reported metrics." (Cxvg) "The experimental setup of the paper is solid." (7smn)

We appreciate the insightful comments and suggestions from the reviewers, and provide additional experimental results and clarifications during the rebuttal phase. Here we summarize them below.

- **Additional Results:** We conducted an ablation study against an axis–angle–based counterpart to highlight the importance of unit quaternion modeling, and provided further analysis on loss weights (7ek4). We reported detailed statistics for the structure prediction results to clarify our superiority over baselines (hsw3). We analyzed the novelty and uniqueness of the de novo generation results via MOFid, and designed a CFG-based method to control the distribution of generated samples toward desired properties (Cxvg). We also reported the time efficiency and parameter numbers of our method and baselines for clearer comparison (C95o).

- **Clarifications:** We clarified the technical contributions (7ek4) and potential in-domain impact (hsw3) of MOF-BFN, explained the architecture design (C95o), discussed limitations of the current framework (Cxvg, C59o), and outlined promising future directions such as DFT in-loop (hsw3) and text-guided generation (7smn).

We believe these additions have improved the clarity and completeness of our work, and we will incorporate them into the revised version.

Best Regards,

Authors

---

### Decision · Program_Chairs · 2025-09-17

**Decision:**

Accept (poster)

**Comment:**

The paper introduces MOF-BFN, a generative model for metal-organic framework (MOF) structure prediction based on Bayesian Flow Networks (BFNs). Its key contributions lie in representing block positions in fractional coordinates to capture periodicity and modeling orientations with unit quaternions based on proposing Bingham BFN. The method jointly predicts lattice parameters, positions, and orientations, achieving state-of-the-art performance in structure prediction, geometric property evaluation, and de novo generation.

The main concerns raised were (a) limited ablation studies and (b) questions about the broader applicability and significance for MOF researchers. The authors have resolved these concerns in the rebuttal and the reviewers support acceptance of this paper.

Overall, I recommend acceptance for this paper. It makes a clear methodological advancement and demonstrates strong experimental evidence.